# Beyond Binary: Continuous State Optimization with Graph-Structured Objectives

**Corinna Cortes** [1]  **Yishay Mansour** [1 2]  **Mehryar Mohri** [1 3]

## Abstract

Large-scale learning systems often face the challenge of balancing multiple, potentially competing objectives, such as fairness, accuracy, and latency. While recent work has formalized this as an optimization problem over binary states, many real-world control parameters—such as fairness thresholds, diversity mixing rates, or resource budgets—are continuous. In this work, we extend the framework to *continuous state spaces*. We model the problem as minimizing a sum of linear objectives subject to *movement costs* that penalize system instability. We capture the local structure of the objectives using a *dependency graph* (or factor graph), where each objective is determined by a subset of the state attributes. To address the tension between exploration and stability, we propose *Lazy Graph-LinUCB*, an algorithm that performs lazy updates to minimize switching costs while maintaining near-optimal regret. Beyond stability, we introduce three advanced mechanisms to exploit the underlying graph structure: (1) an *asynchronous* update schedule that eliminates synchronization overhead in sparse graphs; (2) an *adaptive* algorithm that learns the graph structure from data; and (3) a *joint estimator* that leverages data sharing among correlated objectives to significantly tighten regret bounds. Empirically, we demonstrate that these structural exploitations reduce movement costs by more than a factor of three in heterogeneous systems while maintaining similar cumulative losses.

[1]Google Research, New York, NY; [2]Tel Aviv University, Israel [3]Courant Institute of Mathematical Sciences, New York, NY. Correspondence to: Corinna Cortes <corinna@google.com>, Yishay Mansour <mansour@google.com>, Mehryar Mohri <mohri@google.com>.

*Proceedings of the 43rd International Conference on Machine Learning*, Seoul, South Korea. PMLR 306, 2026. Copyright 2026 by the author(s).

## 1. Introduction

Modern machine learning systems are rarely optimized for a single objective. A recommender system, for instance, must balance relevance against diversity, revenue, latency, and various fairness constraints across demographic groups. These objectives are often inherently competing; increasing the diversity of recommendations may degrade relevance, while enforcing strict fairness constraints may increase latency or reduce revenue. Addressing these conflicts requires a framework for navigating the trade-offs in a data-driven manner. Recent work by Awasthi, Cortes, Mansour, and Mohri (2024; 2026) proposed a model for optimizing such systems based on user feedback (complaints). In their model, the system configuration is represented as a state in a Markov Decision Process (MDP), and the goal is to find a configuration that minimizes the aggregate cost-weighted volume of complaints. A key structural insight of their work was the use of a *dependency graph*—encoding which criteria are mutually incompatible—and *correlation sets* to model local dependencies between criteria, allowing for efficient learning even when the number of criteria is large. However, a significant limitation of this prior work is the assumption that system states are *binary*. Each criterion is modeled as either *fixed* (satisfied) or *unfixed* (violated). While this simplifies the analysis, it fails to capture the nuance of real-world hyperparameters. For example, fairness constraints typically involve a threshold $\tau \in [0, 1]$ (e.g., the allowable gap in True Positive Rates), where a strict setting yields different feedback dynamics than a loose one. Similarly, diversity mixing rates and resource allocation budgets (e.g., latency targets) are inherently continuous parameters rather than binary switches.

In this paper, we propose a natural and powerful extension of the competing objectives framework to *continuous state spaces*. We replace the binary hypercube $\{0, 1\}^k$ with the continuous domain $[0, 1]^k$. To maintain tractability, we generalize the concept of *Correlation Sets* to *Graph-Structured Linear Function Approximation*. We assume the loss function can be approximated by a linear model $\phi(\mathbf{s})^\top \boldsymbol{\theta}$, where the feature map $\phi(\mathbf{s})$ respects the sparsity structure of the underlying dependency graph. Furthermore, we address a critical aspect of continuous control: stability. In the binary

setting, switching a criterion incurred a fixed *fixing cost*. In the continuous setting, we model this as a *movement cost* proportional to the magnitude of the change $\|\mathbf{s}_t - \mathbf{s}_{t-1}\|$. This introduces a non-trivial trade-off between *exploration* (moving the state to disjoint regions to learn the loss landscape) and *stability* (minimizing the cost of adjustment). Our work connects the literature on multi-objective optimization with *Linear Bandits* (Abbasi-Yadkori et al., 2011) and *Bandits with Switching Costs* (Cesa-Bianchi et al., 2013; Dekel et al., 2014; Arora et al., 2019; Amir et al., 2022), offering a unified theoretical framework for optimizing complex, continuous parameter spaces under competing feedback. While our algorithms build on established foundations—specifically the OFUL principle (Optimism in the Face of Uncertainty for Linear bandits) (Abbasi-Yadkori et al., 2011) for bandits and Online Gradient Descent for adversarial settings—our primary algorithmic contribution lies in adapting these mechanisms to the graph-structured and sparsity-constrained nature of the competing objectives problem. In the stochastic setting, LAZYGRAPHLINUCB departs from standard lazy bandit algorithms by leveraging a decomposed determinant trigger. While the base algorithm uses synchronized updates, we also introduce an asynchronous variant that monitors local information gain across the dependency graph, ensuring that stability in one objective does not unnecessarily hinder exploration in disjoint graph neighborhoods. In the adversarial setting, we re-purpose the *virtual iterate* technique not to minimize movement costs (which standard OGD already handles), but to explicitly satisfy operational sparsity budgets, quantifying the precise regret trade-off required to maintain a low-frequency update schedule.

**Contributions.** We make four primary contributions. First, we extend the feedback-driven competing objectives framework from binary system configurations to *continuous state spaces*, modeling the problem as a graph-structured linear bandit with movement costs. This allows us to capture realistic control parameters such as fairness thresholds and resource budgets while rigorously accounting for stability. Second, we propose LAZYGRAPHLINUCB, a stochastic algorithm that uses a decomposed determinant trigger to minimize switching costs while maintaining optimal prediction regret. Third, we introduce three advanced mechanisms to exploit the underlying graph structure: an *asynchronous* update schedule that eliminates synchronization penalties in sparse graphs; an *adaptive* routine that learns dependencies from data; and a *joint estimator* based on a Factor Graph decomposition that tightens regret bounds in dense, correlated systems. Finally, we provide matching minimax lower bounds and empirically demonstrate that our structural exploitations reduce movement costs by more than a factor of three in heterogeneous graph systems, and up to a factor of five in real-world single-objective tasks.

The rest of the paper is organized as follows. Section 2 discusses related work. Section 3 presents our continuous-state formulation. Section 4 introduces the LAZYGRAPHLIN-UCB algorithm and its regret analysis. Section 5 presents advanced strategies for exploiting graph structure, including asynchronous updates and adaptive learning. Section 6 outlines the extension to the adversarial setting. Section 7 provides numerical illustrations to validate the theoretical findings.

## 2. Related Work

Our work sits at the intersection of three lines of research: competing objectives optimization, linear bandits, and online optimization with movement costs.

**Competing objectives and feedback-driven optimization.** Awasthi, Cortes, Mansour, and Mohri (2024; 2026) introduced a feedback-driven framework for optimizing multiple competing objectives, modeled as an MDP over binary states using incompatibility graphs and correlation sets. Our work extends this framework to *continuous* state spaces, replacing binary fixes with smooth parameter tuning. This extension is motivated by the vast literature on multi-objective optimization (Sener & Koltun, 2018; Shah & Ghahramani, 2016; Marler & Arora, 2004) and by algorithmic fairness, where criteria such as equal opportunity and calibration are inherently in tension (Kleinberg et al., 2017; Hardt et al., 2016) and fairness thresholds are continuous parameters (Agarwal et al., 2018; Cotter et al., 2019a). Unlike prior work on fairness in bandits (Joseph et al., 2016; Gillen et al., 2018; Liu et al., 2017) and on agnostic multi-objective algorithms (Cortes et al., 2020), our framework addresses stability explicitly via movement costs and exploits graph sparsity to scale to many criteria.

**Linear bandits.** Our stochastic analysis builds on the OFUL framework of Abbasi-Yadkori et al. (2011) and the contextual bandit work of Li et al. (2010). Graph-structured bandits have been studied in the feedback-graph setting (Alon et al., 2015; Arora et al., 2019), where the graph determines which arm rewards are observed. Our dependency graph plays a different, complementary role: it defines the *feature structure* of the loss function, enabling decomposed estimation (akin to spectral bandits (Valko et al., 2014) or high-dimensional bandit methods with compatibility conditions (Bastani & Bayati, 2020)) and localized policy updates.

**Bandits and online learning with switching costs.** The problem of minimizing regret while limiting the frequency of state changes has been studied for adversarial and stochastic bandits (Dekel et al., 2014; Amir et al., 2022; Cesa-Bianchi et al., 2013; Sherman & Koren, 2021). These works bound the *number* of switches; our movement cost instead penalizes the *magnitude* of each change, capturing the physi-

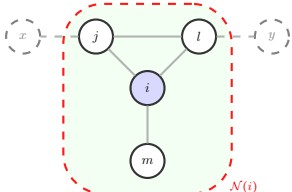

*Figure 1.* Illustration of Graph-Structured Loss. The shaded region $\mathcal{N}(i)$ represents the local neighborhood of criterion $i$. The loss $\ell_{t,i}$ depends strictly on the state parameters of nodes within this region $(\mathbf{s}|_{\mathcal{N}(i)})$. Changes to distant, non-neighboring nodes (such as $x$ and $y$) do not affect the gradient for criterion $i$, enabling LAZY-GRAPHLINUCB to decompose the global optimization problem into local sub-problems.

cal reality of continuous-parameter adjustment. In the online convex optimization literature, smoothed or lazy variants have been studied for switching costs (Chen et al., 2018; Zhang et al., 2021; Goel et al., 2019; Andrew et al., 2013; Kalai & Vempala, 2005). Our Randomized Lazy OGD (Section 6) connects to this line of work, while our stochastic LAZYGRAPHLINUCB achieves *data-dependent* laziness via the determinant-doubling trigger, avoiding the fixed-probability randomization required in adversarial settings.

An extended discussion of related work, covering the broader fairness and multi-criteria optimization literature in greater depth, is given in Appendix A.

## 3. Problem Setup

**Notation.** We denote the set of criteria (nodes) by $\mathcal{V} = \{1, \dots, k\}$. The system state is a vector $\mathbf{s} \in \mathcal{S} = [0,1]^k$. We assume an underlying dependency graph $\mathcal{G} = (\mathcal{V}, E)$, where $\mathcal{N}(i)$ denotes the neighborhood of $i$; see Figure 1 for an example.

**Graph-Structured Loss.** At round $t$, the learner selects $\mathbf{s}_t$ and observes a loss vector $\ell_t(\mathbf{s}_t) \in \mathbb{R}^k$. We assume the expected loss $\mu_i(\mathbf{s}) = \mathbb{E}[\ell_{t,i}(\mathbf{s})]$ depends only on the local neighborhood $\mathcal{N}(i)$. Specifically, we adopt a linear function approximation:

$$\mu_i(\mathbf{s}) = \left\langle \phi_i(\mathbf{s}|_{\mathcal{N}(i)}), \boldsymbol{\theta}_i^* \right\rangle, \tag{1}$$

where $\phi_i : [0,1]^{|\mathcal{N}(i)|} \to \mathbb{R}^{d_i}$ is a local feature map and $\boldsymbol{\theta}_i^* \in \mathbb{R}^{d_i}$ is an unknown parameter. This generalizes the *Correlation Sets* of Awasthi et al. (2024) to continuous domains. The total loss is $L(\mathbf{s}) = \sum_{i=1}^k \mu_i(\mathbf{s})$.

**Movement Costs and Regret.** Unlike standard bandits, changing the state is costly. We impose a movement penalty $\lambda \|\mathbf{s}_t - \mathbf{s}_{t-1}\|_1$. The goal is to minimize cumulative regret against the optimal static state $\mathbf{s}^*$:

$$\mathrm{Reg}_T = \sum_{t=1}^T \left( L(\mathbf{s}_t) - L(\mathbf{s}^*) \right) + \sum_{t=1}^T \lambda \|\mathbf{s}_t - \mathbf{s}_{t-1}\|_1. \tag{2}$$

**Background: LinUCB.** Our approach builds on the OFUL principle (Abbasi-Yadkori et al., 2011). For a standard linear bandit, one maintains a regularized least-squares estimator $\widehat{\theta}_t = \mathbf{V}_t^{-1} \sum_{\tau=1}^t x_\tau y_\tau$ with covariance $\mathbf{V}_t = \lambda_{\mathrm{reg}} \mathbf{I} + \sum_{\tau=1}^t x_\tau x_\tau^\top$, where $\lambda_{\mathrm{reg}}$ is a positive initialization hyperparameter. The confidence set is $\mathcal{C}_t = \{\theta : \|\theta - \widehat{\theta}_t\|_{\mathbf{V}_t} \le \beta_t\}$, where $\beta_t = \widetilde{O}(\sqrt{d \log t})$ is the confidence radius (see Lemma 12). We adapt this framework to the multi-objective setting by maintaining $k$ local estimators.

## 4. Algorithm: Lazy Graph-LinUCB

We propose an algorithm, LAZYGRAPHLINUCB, that efficiently handles the trade-off between exploration and movement costs by using a lazy update schedule. The algorithm maintains separate linear estimators for each criterion but coordinates their updates to minimize global movement.

### 4.1. Algorithm Description

The algorithm, LAZYGRAPHLINUCB (Algorithm 1), extends the standard LinUCB framework to the loss minimization setting. While the original algorithm was named for the *Upper Confidence Bound* on rewards, we retain the name to denote the family of algorithms based on optimism in the face of uncertainty. In our minimization context, optimism corresponds to using a *Lower Confidence Bound (LCB)*. The algorithm operates as follows:

1. Local Estimation: For each criterion $i \in [k]$, maintain a regularized least-squares estimator for $\boldsymbol{\theta}_i^*$ using only the features $\phi_i(\mathbf{s})$. Let $\mathbf{V}_{t,i}$ be the covariance matrix and $\widehat{\boldsymbol{\theta}}_{t,i}$ the estimate at time $t$.

2. Global Optimistic Bound: Construct a global score for the loss function $L(\mathbf{s})$. To ensure exploration, we use the Lower Confidence Bound (LCB), defined as the estimated loss minus the exploration bonus:

$$\mathrm{LCB}_t(\mathbf{s}) = \sum_{i=1}^k \left( \langle \phi_i(\mathbf{s}), \widehat{\boldsymbol{\theta}}_{t,i} \rangle - \beta_{t,i} \|\phi_i(\mathbf{s})\|_{\mathbf{V}_{t,i}^{-1}} \right). \tag{3}$$

3. Lazy Updates: The algorithm maintains an active state $\mathbf{s}_{\mathrm{active}}$. It only re-solves the optimization problem when the determinant of the covariance matrix for *any* criterion has doubled significantly since the last update. If $\max_i \det(\mathbf{V}_{t,i}) > 2 \det(\mathbf{V}_{\mathrm{last},i})$, then

$$\mathbf{s}_{\mathrm{active}} \leftarrow \underset{\mathbf{s} \in \mathcal{S}}{\arg\min} \, \mathrm{LCB}_t(\mathbf{s}). \tag{4}$$

Note that the optimization in (4) is non-convex in general. However, in many practical settings (e.g., linear or quadratic bases over convex domains), this optimization is convex or admits efficient approximate solvers.

**Standard LinUCB and Movement Costs.** One might ask why the standard LinUCB algorithm cannot be applied

**Algorithm 1** Synchronous Lazy Graph-LinUCB

1: **Input:** Dependency Graph $\mathcal{G}$, Regularization $\lambda_{\text{reg}}$, Confidence $\delta$.
2: **Initialize:** For all $i \in [k]$: $\mathbf{V}_{0,i} = \lambda_{\text{reg}}\mathbf{I}_{d_i}$, $\widehat{\boldsymbol{\theta}}_{0,i} = \mathbf{0}$, $\tau_i = 0$.
3: **Initialize:** Active state $\mathbf{s}_{\text{active}} \in \mathcal{S}$ (arbitrary).
4: **for** $t = 1$ to $T$ **do**
5:     **Check Trigger:**
6:     **if** $\exists i \in [k]$ such that $\det(\mathbf{V}_{t-1,i}) > 2\det(\mathbf{V}_{\tau_i,i})$ **then**
7:         *// Sufficient information gained; update active policy*
8:         Calculate LCB score: $\text{LCB}_t(\mathbf{s}) = \sum_{j=1}^{k}(\langle\phi_j(\mathbf{s}), \widehat{\boldsymbol{\theta}}_{t-1,j}\rangle - \beta_{t-1,j}\|\phi_j(\mathbf{s})\|_{\mathbf{V}_{t-1,j}^{-1}})$
9:         Update Policy: $\mathbf{s}_{\text{active}} \leftarrow \arg\min_{\mathbf{s}\in\mathcal{S}} \text{LCB}_t(\mathbf{s})$
10:        Update Last-Sync Times: $\tau_j \leftarrow t-1$ for all $j \in [k]$
11:     **end if**
12:     **Play:** $\mathbf{s}_t = \mathbf{s}_{\text{active}}$
13:     **Observe:** Loss vector $\ell_t(\mathbf{s}_t)$
14:     **Update Estimates:**
15:     **for** $i = 1$ to $k$ **do**
16:         $\mathbf{V}_{t,i} = \mathbf{V}_{t-1,i} + \phi_i(\mathbf{s}_t)\phi_i(\mathbf{s}_t)^{\top}$
17:         $\widehat{\boldsymbol{\theta}}_{t,i} = \mathbf{V}_{t,i}^{-1}\sum_{\tau=1}^{t}\phi_i(\mathbf{s}_\tau)\ell_{\tau,i}$
18:     **end for**
19: **end for**

---

directly to this setting. Recall that at every round $t$, standard LinUCB selects the state $\mathbf{s}_t = \arg\min_{\mathbf{s}\in\mathcal{S}}\text{LCB}_t(\mathbf{s})$. Since the confidence intervals and estimated parameters $\widehat{\boldsymbol{\theta}}_t$ evolve stochastically with every new observation, the global minimizer of the LCB score can oscillate rapidly. Consequently, standard LinUCB can incur a movement cost of $\Omega(T)$, yielding linear total regret. To address this, our LAZYGRAPHLINUCB algorithm freezes the active policy, updating it only when the information gain (measured by the determinant of the covariance matrix) is sufficient to justify the cost of switching.

### 4.2. Theoretical Analysis

We now derive the regret bound for LAZYGRAPHLINUCB. The analysis relies on three key lemmas: establishing confidence ellipsoids, bounding the number of policy updates (epochs), and bounding the prediction error incurred during the *lazy* frozen periods.

#### 4.2.1. KEY LEMMAS

**Confidence Sets.** We define the confidence sets using the standard self-normalized martingale bounds. For a criterion $i$, define the regularized least-squares estimator $\widehat{\boldsymbol{\theta}}_{t,i} = \mathbf{V}_{t,i}^{-1}\sum_{\tau=1}^{t}\phi_i(\mathbf{s}_\tau)y_{\tau,i}$.

To analyze the regret, we rely on standard self-normalized

martingale bounds (Abbasi-Yadkori et al., 2011) to define confidence ellipsoids $\mathcal{C}_t$ centered at $\widehat{\boldsymbol{\theta}}_t$ (see Lemma 12 in Appendix E).

Let $\mathcal{E}$ denote the *good event* that the true parameter $\boldsymbol{\theta}_i^*$ lies within the confidence ellipsoid $\mathcal{C}_{t,i}$ for all criteria $i \in [k]$ and all rounds $t \geq 1$. By a union bound over all $i$ and $t$, $\mathbb{P}(\mathcal{E}) \geq 1 - \delta$. We condition the rest of the analysis on $\mathcal{E}$.

**Movement Bounds.** The algorithm updates $\mathbf{s}_{\text{active}}$ only when the determinant of the covariance matrix doubles. Let $\tau_0, \ldots, \tau_M$ be the update times. We call $[\tau_j, \tau_{j+1} - 1]$ the $j$-th *epoch*.

**Lemma 1** (Bound on Number of Epochs). *The total number of updates $M$ is bounded by:*

$$M \leq \sum_{i=1}^{k} d_i \log_2\left(1 + \frac{TL^2}{d_i\lambda_{\text{reg}}}\right).$$

**Prediction Error.** A key challenge in lazy algorithms is that we play an action $\mathbf{s}_t$ based on a *stale* covariance matrix $\mathbf{V}_{\tau(t)}$. We must bound the error this staleness introduces.

**Lemma 2** (Bounded Error per Epoch). *Under the feature normalization assumption $\|\phi\|_{\mathbf{V}^{-1}} \leq 1$, for any epoch $j$, the sum of squared stale norms is bounded by:*

$$S_j = \sum_{t=\tau_j}^{\tau_{j+1}-1} \|\phi(\mathbf{s}_t)\|_{\mathbf{V}_{\tau_j}^{-1}}^2 \leq 3.$$

The proofs of the lemmas are given in Appendix E.

#### 4.2.2. MAIN REGRET GUARANTEE

We can now combine these ingredients to bound the total regret.

**Theorem 3** (Regret of Lazy Graph-LinUCB). *Assume the movement cost is bounded by $\lambda$ per switch. Let $d_i$ be the dimension of the feature map for criterion $i$, and $d_{\max} = \max_i d_i$. The cumulative regret of LAZYGRAPHLINUCB is bounded by:*

$$\text{Reg}_T \leq \widetilde{O}\left(\sqrt{T}\sum_{i=1}^{k} d_i + \lambda k^2 d_{\max}\log T\right). \tag{5}$$

*Proof Sketch.* The total regret decomposes into *Movement* and *Prediction* regret. For movement, we bound the number of updates $M$ by $O(d_{\max}k\log T)$ using a determinant-doubling argument (Lemma 1). Since each switch costs at most $\lambda k$, the movement regret is logarithmic. For prediction, we bound the instantaneous regret using the confidence width evaluated at the *stale* covariance matrix. A crucial step is showing that the sum of squared *stale norms* within any lazy epoch is bounded by a constant ($S_j \leq 3$) despite the staleness (Lemma 2). Summing over all epochs yields a prediction regret scaling with $\sqrt{T}\sum d_i$. (See Appendix E.4 for full derivation). $\square$

**Corollary 4** (Robustness to Graph Misspecification). *Suppose the learner operates using a proxy graph $\mathcal{G}'$ such that $E \subseteq E'$ (a super-graph). Let $d'_i$ denote the feature dimension induced by $\mathcal{G}'$. Then,* LAZYGRAPHLINUCB *remains valid and guarantees a regret bound scaling with the proxy dimensions:*

$$\mathsf{Reg}_T \le \widetilde{O}\left(\sqrt{T}\sum_{i=1}^{k} d'_i + \lambda k^2 d'_{\max}\log T\right). \quad (6)$$

### 4.3. Minimax Lower Bound

We now present a lower bound demonstrating that the dependency on the sum of dimensions in our upper bound is tight.

**Theorem 5** (Minimax Lower Bound). *For any learning algorithm, there exists a graph structure $\mathcal{G}$ and a sequence of loss functions such that the expected regret is lower bounded by:*

$$\mathbb{E}[\mathsf{Reg}_T] = \Omega\left(\sqrt{T}\sum_{i=1}^{k} d_i\right). \quad (7)$$

**Comparison to Naive Reduction.** One might seek to solve this problem by scalarizing the objective $L(\mathbf{s}) = \sum \ell_i(\mathbf{s})$ and applying a standard high-dimensional linear bandit algorithm to the concatenated parameter $\boldsymbol{\theta}^* \in \mathbb{R}^{\sum d_i}$. While the dimension dependence ($\sum d_i$) would match our result, this naive reduction fails on two fronts. First, *Feedback Granularity*: By observing only the scalar sum, the learner faces an aggregated noise variance of $\sigma_{\text{agg}}^2 = \sum_{i=1}^{k}\sigma_i^2 \approx k$. This inflates the regret by a factor of $\sqrt{k}$ compared to our component-wise approach which exploits the vector-valued feedback. Second, and most importantly, *Stability*: The global approach cannot distinguish which local component caused a loss increase. Consequently, it updates the entire parameter vector at once, incurring a movement cost proportional to the global dimension $k$ at every switch, whereas our approach restricts movement costs to local neighborhoods.

## 5. Exploiting Graph Structure

While the global Lazy strategy (Section 4) achieves optimal prediction regret, it does not fully leverage the sparsity and correlations inherent in the dependency graph $\mathcal{G}$. In this section, we present three key extensions that exploit this structure: an *asynchronous* update schedule that minimizes movement costs in sparse graphs, an *adaptive* procedure that learns the graph structure when it is initially unknown, and a *factor-graph* decomposition that leverages data sharing among correlated objectives to tighten regret bounds. Table 1 summarizes when each variant is best suited and what additional assumptions it requires.

All detailed proofs are deferred to Appendix E.

*Table 1.* **Algorithm Variant Selection Guide.** Summary of when to use each variant and what additional assumptions it requires.

| Variant | Best When | Key Assumption |
|---|---|---|
| Global Lazy (Section 4) | Dense graph or synchronized updates needed | None beyond standard linear bandit |
| Async Lazy (Section 5.1) | Sparse/heterogeneous graph; criteria learn at different rates | Assumption A: LCB objective is PL (satisfied for large $\lambda_{\text{reg}}$) |
| Adaptive Refinement (Section 5.2) | Graph structure completely unknown | Assumptions B (signal strength) and C (diverse contexts) |
| Joint Estimator (Section 5.3) | Dense graph with correlated objectives sharing latent factors | Assumption D: factor decomposability |

### 5.1. Asynchronous Updates for Sparse Graphs

In the LAZYGRAPHLINUCB algorithm described above, the update condition is *global*: if any single criterion triggers a determinant doubling, the entire policy vector $\mathbf{s}_{\text{active}}$ is re-optimized. While this guarantees regret optimality in the worst case (dense graphs), it may be inefficient for systems with sparse dependency graphs. For instance, learning new information about a latency criterion should ideally not force a reconfiguration of a graph-distant fairness criterion. To exploit sparsity, we propose an asynchronous algorithm, ASYNC-LAZYGRAPHLINUCB, which performs *local* policy updates. Such a solution is necessary, as global lazy updates are suboptimal for sparse graphs.

**Mechanism.** The algorithm maintains the same local estimators but modifies the update schedule:

1. Local Trigger: Each criterion $i$ maintains its own last-update time $\tau_i$. An update is triggered for criterion $i$ only if $\det(\mathbf{V}_{t,i}) > 2\det(\mathbf{V}_{\tau_i,i})$.

2. Local Update: When criterion $i$ triggers, we solve for the new active policy $\mathbf{s}_{\text{active}}$ by optimizing only the variables in the local neighborhood $\mathcal{N}(i)$, keeping all variables $s_j$ for $j \notin \mathcal{N}(i)$ fixed at their previous values.

*Tie-Breaking:* In the event that multiple criteria trigger simultaneously (i.e., $|\mathcal{A}_t| > 1$), we perform a synchronized update over the union of their neighborhoods $\bigcup_{i\in\mathcal{A}_t}\mathcal{N}(i)$. This ensures that coupled dependencies between simultaneously triggering nodes are resolved jointly.

3. Synchronization: The time $\tau_i$ is updated to $t$. Note that because neighborhoods overlap, an update to $\mathcal{N}(i)$ may effectively update partial state for neighbors $j \in \mathcal{N}(i)$, but it does not propagate to the entire graph.

**Assumption A: BCD Convergence.** We assume the objective function is sufficiently regular—specifically, that it satisfies the Polyak-Łojasiewicz (PL) condition—such that the Block Coordinate Descent (BCD) updates in the asynchronous procedure converge to the global minimizer with negligible optimality gap.

*When does Assumption A hold?* The LCB objective $f_t(\mathbf{s}) = \sum_i (\langle \phi_i, \widehat{\boldsymbol{\theta}}_{t,i} \rangle - \beta_{t,i} \|\phi_i(\mathbf{s})\|_{\mathbf{V}_{t,i}^{-1}})$ is the sum of a *strongly convex* empirical loss term (the regularizer contributes $\lambda_{\text{reg}} \|\mathbf{s}\|^2 / 2$ per coordinate) and a concave correction (the exploration bonus). For any smooth, bounded feature map $\phi_i$ on a compact domain, the Hessian of the exploration bonus is bounded in operator norm by some constant $C(\phi_i)$ that depends only on the Lipschitz constant of $\phi_i$ and the problem geometry. Consequently, whenever $\lambda_{\text{reg}} > C(\phi_i)$, the net Hessian is positive definite and the LCB objective is $(\lambda_{\text{reg}} - C(\phi_i))$-strongly convex, hence PL (see Lemma 13 in Appendix E). In particular, as the exploration bonus shrinks over time ($\beta_{t,i} \to 0$), this regime is automatically entered. The algorithm is designed to operate in this regularity regime by an appropriate choice of $\lambda_{\text{reg}}$.

**Theoretical Improvement.** This asynchronous strategy yields a significantly tighter bound on the movement cost for sparse graphs. In the global strategy, every update incurs a movement cost bounded by the total dimension $k$ (since all $s_i$ might shift). In the asynchronous strategy, an update initiated by criterion $i$ only shifts coordinates within $\mathcal{N}(i)$, bounding the movement cost by the local degree $|\mathcal{N}(i)|$.

**Theorem 6** (Regret of Asynchronous Lazy Graph-LinUCB). *Let $\Delta = \max_i |\mathcal{N}(i)|$ be the maximum degree of the dependency graph. In the regularity regime where $\lambda_{\text{reg}}$ is chosen sufficiently large to satisfy Assumption A (see Lemma 13), the cumulative regret of the asynchronous algorithm is bounded by:*

$$\mathsf{Reg}_T \le \widetilde{O}\left(\sqrt{T} \sum_{i=1}^{k} d_i + \lambda \Delta d_{\max} k \log T\right). \tag{8}$$

*Proof Sketch.* The analysis applies the global strategy locally. The key challenge is that feature vectors $\phi_i(\mathbf{s}_t)$ vary during an epoch due to neighbor updates. We prove that a similar trace-determinant bound (that of Lemma 2) holds, modular a small constant factor, even for time-varying features, ensuring the prediction regret remains optimal. For movement costs, since an update is restricted to the local neighborhood $\mathcal{N}(i)$, the cost per switch is bounded by $\lambda \Delta$ rather than $\lambda k$. (See details in Appendix E.8). $\square$

This result highlights that for sparse graphs (where $\Delta \ll k$), the asynchronous strategy effectively decouples the learning problems, reducing the dependency on the global dimension $k$ in the movement term. Note that determinant-based triggers are a standard tool for computational efficiency in linear bandits; we repurpose them here for stability. Crucially, a naive application to graph-structured problems leads to a synchronization penalty, where stable criteria update unnecessarily. Our primary contribution is the asynchronous analysis (Theorem 6), which proves that decomposing the

trigger conditions is essential for optimal movement costs in heterogeneous systems.

## 5.2. Adaptive Graph Learning

In many applications, the true dependency graph $\mathcal{G}$ may be unknown. While relying on a safe super-graph ensures validity (Corollary 4), it incurs suboptimal movement costs. Here, we show that when the true dependencies satisfy a minimum signal strength condition and the contexts are sufficiently diverse, similar to the *compatibility conditions* required for high-dimensional bandits (Bastani & Bayati, 2020), the learner can adaptively recover the sparse graph structure.

The adaptive graph learning mechanism in this section is primarily a theoretical guarantee showing that a priori knowledge of the dependency graph is not strictly necessary for optimal regret. Our goal is not to propose a *turnkey algorithm* for structure learning under movement costs, but to establish that, under standard identifiability conditions, the learner can provably recover and exploit sparsity without compromising regret rates. Practical variants and empirical evaluation are left to future work.

**Assumption B: Minimum Signal Strength.** Let $\phi_{\text{full}}(\mathbf{s})$ be the feature map corresponding to the complete graph. We assume the true parameter $\boldsymbol{\theta}_{i,\text{full}}^*$ is supported on a sparse subset of indices corresponding to the true neighborhood $\mathcal{N}(i)$. For any active neighbor $j \in \mathcal{N}(i)$, the dependency is bounded away from zero: $|\theta_{i,j}^*| \ge \gamma > 0$.

**Assumption C: Diverse Contexts.** During the warm-up phase, we assume the system visits a diverse set of states such that the minimum eigenvalue of the covariance matrix grows linearly. Formally, there exists $\kappa > 0$ such that for any $\tau$, $\lambda_{\min}(\mathbf{V}_{\tau,i}) \ge \kappa \tau$. This can be enforced by adding isotropic noise to the actions during the warm-up phase.

**Algorithm: Adaptive Graph Refinement.** We propose a two-phase strategy:

1. Warm-up Phase: Initialize with the complete graph $\mathcal{G}_{\text{dense}}$ with dimension $d_{\text{total}}$. Run LAZYGRAPHLINUCB for $\tau$ rounds.

2. Sparsity Test: At $t = \tau$, for each criterion $i$, compute the estimator $\widehat{\boldsymbol{\theta}}_{\tau,i}$ using the dense covariance matrix. Construct the estimated neighborhood $\widehat{\mathcal{N}}(i) = \{j : |\widehat{\theta}_{\tau,i,j}| > \gamma/2\}$.

3. Refinement: Update the graph to $\widehat{\mathcal{G}} = (\mathcal{V}, \widehat{E})$ based on these neighborhoods. For $t > \tau$, resume LAZYGRAPHLIN-UCB using $\widehat{\mathcal{G}}$ (retaining the covariance matrices $\mathbf{V}_{\tau,i}$).

**Theorem 7** (Regret of Adaptive Graph Refinement). *Let $\mathcal{G}_{\text{dense}}$ be the initial dense graph and $\mathcal{G}$ be the true sparse graph. Set the warm-up length to: $\tau = \frac{16\beta_T^2}{\kappa\gamma^2}$, where $\beta_T$ is the confidence radius for dimension $d_{\text{total}}$ as defined in*

*Lemma 12. Under Assumptions B and C, with probability at least $1 - \delta$, the algorithm recovers the true graph $\widehat{\mathcal{G}} = \mathcal{G}$, and the total regret is bounded by:*

$$\mathsf{Reg}_T \le \widetilde{O}\left( \frac{d_{\text{total}}^{1.5}}{\kappa\gamma^2} + \sqrt{T}\sum_{i=1}^{k} d_i + d_{\text{total}}^2 \log T \right). \quad (9)$$

*Proof Sketch.* The proof relies on the existence of a *Warm-up* phase of length $\tau$. We show that with high probability, the estimator $\widehat{\theta}_\tau$ calculated on the dense graph correctly identifies the support of the true parameter, provided the signal strength $\gamma$ is sufficient (Assumptions B and C). Once the graph structure is recovered ($\widehat{\mathcal{G}} = \mathcal{G}$), the regret bound for the remaining $T - \tau$ rounds follows from Theorem 3. (See Appendix E.9 for full derivation.) $\qquad\square$

**Regime of Validity.** The regret bound consists of a constant *learning cost* (scaling with $\gamma^{-2}$) and a term scaling with $\sqrt{T}$. This bound is meaningful when the horizon is sufficiently long to amortize the warm-up phase, specifically when $T \gg \widetilde{\Omega}(\gamma^{-4})$. In the regime of extremely weak signals ($\gamma \le T^{-1/4}$), the cost of distinguishing the true support outweighs the benefits of sparsity within the horizon $T$. In such cases, the learner is better off defaulting to the conservative super-graph strategy (Corollary 4), effectively treating weak signals as noise.

### 5.3. Data Sharing for Correlated Objectives

The regret bounds in Section 4 scale with $\sum_{i=1}^{k} d_i$, treating each criterion as an independent learning task. However, the graph structure often implies stronger correlations: neighbors $i$ and $j$ in the dependency graph typically depend on the same underlying state variables or latent factors. In this section, we show that by formalizing the global structure explicitly as a *Factor Graph* decomposition, a standard realistic assumption in graphical models (see e.g., (Cortes et al., 2016)), we can further reduce the regret.

**Assumption D: Factor Decomposability.** We assume the global loss function $L(\mathbf{s})$ decomposes over the maximal cliques $\mathcal{C}$ of the graph $\mathcal{G}$:

$$L(\mathbf{s}) = \sum_{C \in \mathcal{C}} \langle \psi_C(\mathbf{s}|_C), \mathbf{w}_C^* \rangle, \quad (10)$$

where $\psi_C$ are local feature maps for each clique and $\mathbf{w}_C^* \in \mathbb{R}^{d_C}$ are unique clique parameters. This differs from the node-wise assumption in (1) ($\mu_i(\mathbf{s}) = \langle\phi_i, \theta_i^*\rangle$). There, $\theta_i^*$ aggregates all factors affecting node $i$. Here, we disentangle them. For instance, if $i$ and $j$ share a clique $C$, they share the parameter $\mathbf{w}_C^*$. This is physically realistic in systems where constraints share underlying physics (e.g., two fairness constraints depending on the same sensitive attribute coefficients).

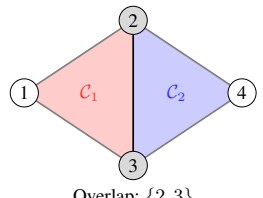

Overlap: $\{2, 3\}$

*Figure 2.* **Factor Graph Decomposition.** Illustration of two overlapping cliques. Nodes 2 and 3 participate in both interaction $\mathcal{C}_1$ and $\mathcal{C}_2$. While a node-wise estimator treats these dependencies independently (counting the shared dimension twice), the Joint Estimator exploits the overlap to learn unique parameters for each clique, reducing the effective dimension $d_{\text{eff}}$.

**Algorithm: Joint-LazyGraphLinUCB.** We construct a *Joint Estimator* by concatenating all unique clique parameters into a single vector $\theta_{joint}^* \in \mathbb{R}^{d_{\text{eff}}}$, where $d_{\text{eff}} = \sum_{C \in \mathcal{C}} d_C$. At each round $t$, the learner receives a feedback vector $\mathbf{Y}_t = \ell_t(\mathbf{s}_t) \in \mathbb{R}^k$. We treat this as a single linear bandit problem with a *macro-action* $\mathbf{s}_t$ and a vector-valued reward. The estimator minimizes the joint regularized least squares objective. The update rule is global: $\mathbf{s}_{\text{active}}$ is updated only when the determinant of the *joint* covariance matrix doubles.

**Theorem 8** (Regret of Joint Estimator). *Let $d_{\text{eff}} = \sum_{C \in \mathcal{C}} d_C$ be the sum of the dimensions of the maximal cliques. The cumulative regret of the Joint-LazyGraphLinUCB is bounded by:*

$$\mathsf{Reg}_T \le \widetilde{O}\left( d_{\text{eff}}\sqrt{kT} + \lambda k^2 d_{\text{max}} \log T \right). \quad (11)$$

**Comparison to Node-Wise Bound.** The standard bound (Theorem 3) scales with $\sum_{i=1}^{k} d_i$. In the Factor model, the joint estimator scales with $d_{\text{eff}}\sqrt{k} = (\sum_C d_C)\sqrt{k}$. Consider a dense case where $k$ nodes form a single clique of dimension $D$. The Node-wise bound scales as $kD$. The Joint bound scales as $D\sqrt{k}$. Thus, the joint estimator improves the regret by a factor of $\sqrt{k}$ in dense, highly correlated systems, see Figure 2.

## 6. Extension: Adversarial Setting

We also extend our framework to the adversarial full-information setting, where loss functions $\ell_t(\cdot)$ are arbitrary convex functions chosen by an adversary. Unlike the stochastic setting, *smart* determinant-based laziness is not possible. To satisfy a strict switching budget $M$, we propose RANDOMIZED LAZY OGD, which maintains a virtual OGD iterate but updates the deployed state $\mathbf{s}_t$ only with probability $p = M/T$.

We prove this achieves $\mathbb{E}[\mathsf{Reg}_T] \le \tilde{O}(D\,G\,T/\sqrt{M})$, where $D$ is the diameter and $G$ is the Lipschitz constant (Theorem 10), quantifying the *price of sparsity*: regret degrades from $\sqrt{T}$ to $T/\sqrt{M}$. An extension for heterogeneous budgets $\mathbf{M} = (M_1, \ldots, M_k)$ is also provided. Full details are

in Appendix D.

# 7. Numerical Illustrations

To validate our theoretical findings, we conducted two synthetic experiments and one real-world application. Our primary contribution is theoretical; these experiments are not large-scale benchmarks, but numerical illustrations isolating and validating specific mechanisms predicted by our analysis: the reduction of jitter and the elimination of synchronization overhead.

## 7.1. Single-Clique Stability Analysis

We first compared LAZYGRAPHLINUCB against standard LinUCB on a simple $d = 2$ dimensional linear bandit problem over $T = 300$ rounds. The goal was to minimize a noisy linear loss function $\ell_t(s) = \langle s, \theta^* \rangle + \mathcal{N}_t(0, \sigma^2)$. The parameters used were $\lambda_{\text{reg}} = 0.1$, $\beta = 0.5$ for the exploration term, and noise parameter $\sigma = 0.5$. Both algorithms rely on the same regularized least-squares estimator; the critical difference lies in the update schedule. Standard LinUCB updates its policy at every step to the minimizer of the current LCB, whereas LAZYGRAPHLINUCB only updates when the determinant of the covariance matrix doubles. We note that the cumulative prediction regret for both algorithms was statistically indistinguishable, confirming that the stability gains did not come at the cost of accuracy.

**Stability vs. Jitter.** Figure 3 (Left) illustrates the core mechanism of our algorithm. Because the estimator $\widehat{\theta}_t$ fluctuates with every noisy sample, the standard LinUCB action $s_t$ jitters constantly. In contrast, LAZYGRAPHLINUCB freezes the action for stable epochs. The initial fluctuations in the first $\approx 20$ rounds correspond to the rapid shrinkage of the confidence ellipsoid (where the determinant doubles frequently). However, as $t$ increases, the updates become exponentially rare.

**Cost Reduction.** Figure 3 (Right) confirms our regret bounds. The standard algorithm incurs linear movement cost ($\Omega(T)$), making it unusable in systems with high switching penalties. Our lazy algorithm maintains logarithmic movement cost ($O(\log T)$) while provably retaining the optimal $\widetilde{O}(\sqrt{T})$ prediction accuracy. (Note: In our experiments, the cumulative prediction regret of the Lazy algorithm was statistically indistinguishable from the Standard algorithm; we omit the redundant learning curves for brevity.)

## 7.2. Heterogeneous Graph Scalability

To validate the benefits of our graph-structured approach, we simulated a larger system with $k = 10$ independent criteria (cliques), each of dimension $d = 2$. To mirror real-world conditions where different objectives learn at different rates (e.g., a noisy *CTR* objective vs. a stable *Latency* objective),

we introduced heterogeneity by assigning a random stiffness parameter $\lambda_{\text{reg},i} \in [0.01, 2.0]$ to each clique. The higher the $\lambda_{\text{reg},i}$, the longer it will take for a clique to reach its triggering threshold.

The results in Figure 4 highlight the critical importance of the asynchronous extension (Section 5.1). Under GLOBAL LAZY, all cliques are forced to update whenever *any* criterion triggers, so even stiff (slow-learning) cliques update frequently despite gathering little new information. ASYNC LAZY (Theorem 6) eliminates this waste. As shown in Table 2, stable components (e.g., Clique 9) incur a 3.7× waste factor under the global strategy as it is updated 41 times vs. 11 times under the asynchronous strategy; decoupling updates reduces total system-wide moves by isolating changes to the relevant local neighborhood.

*Table 2.* **Mechanism of Efficiency (Excerpt).** Comparison of update counts for representative cliques. Volatile cliques update frequently in both strategies; stable cliques freeze in Async but are forced to move in Global. See Appendix F for the full breakdown of all 10 cliques.

| Clique Type | Stiffness, $\lambda_{\text{reg}}$ | Global, moves | Async, moves | Waste factor |
|---|---|---|---|---|
| Volatile (ID 4) | 0.19 | 41 | 15 | 2.7× |
| Stable (ID 9) | 1.40 | 41 | 11 | **3.7×** |
| **Total (All 10)** | - | **410** | **131** | **3.1×** |

## 7.3. Real-World Data: Fairness Threshold Tuning

To validate our framework in a realistic setting, we applied it to a fairness tuning task using the *Adult Census dataset*. We trained a logistic regression classifier to predict income > \$50K and formulated the problem of tuning the decision threshold $\tau \in [0, 1]$ as a bandit problem with inherent conflicting objectives: maximizing accuracy of the income prediction while minimizing the Equal Opportunity gap. Specifically, we scalarize the global objective as the sum of the error rate and the fairness violation:

$$L(\tau) = (1 - \text{Accuracy}(\tau)) + |\text{TPR}_{\text{male}}(\tau) - \text{TPR}_{\text{female}}(\tau)|.$$

To capture the non-linear dependence of these metrics on the threshold, we use a quadratic feature map $\phi(\tau) = [1, \tau, \tau^2]^\top$. We initialized the threshold at a low value ($\tau = 0.2$) to observe the learning trajectory as the agent discovers the high-threshold region ($\tau \approx 0.9$) required for fairness. We used $\lambda_{\text{reg}} = 5$ to initialize the covariance matrix $\mathbf{V}$. To simulate realistic feedback variability, we added zero-mean Gaussian noise ($\sigma = 0.05$) to the observed losses. While this experiment focuses on a single parameter ($k = 1$) and thus does not exercise the graph sparsity mechanisms (illustrated in Section 7.2), it serves to empirically validate the efficiency and stability of the continuous-state LAZY formulation on a real-

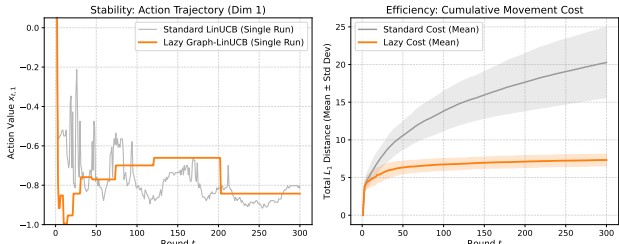

*Figure 3.* **Impact of Lazy Updates.** (Left) The trajectory of the selected action $s_t$ (dimension 1). Standard LinUCB (grey) exhibits constant jitter, while LAZYGRAPHLINUCB (orange) freezes the action. (Right) Cumulative movement cost averaged over 100 runs. The standard algorithm accumulates linear cost ($\approx 22$), while the lazy strategy saturates ($\approx 8$).

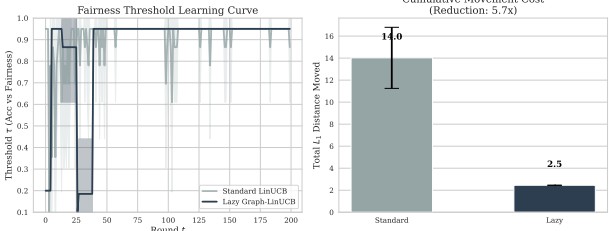

*Figure 5.* **Fairness Threshold Tuning on Adult Dataset.** (Left) Mean learning trajectory ($\pm 1$ std. dev.) over 10 independent trials. The Lazy algorithm (blue) consistently tracks the optimal threshold with significantly reduced variance compared to the standard LinUCB approach (grey). (Right) Cumulative movement cost. The Lazy strategy significantly reduces operational churn, achieving a $\approx 5.7\times$ reduction in total movement cost.

istic, non-convex loss surface, complementing the structural verification in the synthetic experiments.

**Results.** Figure 5 (Left) shows the mean threshold trajectory averaged over 10 random seeds. Both algorithms successfully converge to the fair region ($\tau \approx 0.9$). However, the standard LinUCB algorithm (grey) exhibits high variance and frequent oscillations due to sensitivity to batch noise. In contrast, LAZYGRAPHLINUCB (blue) demonstrates a stable learning curve with tight confidence bands, filtering out transient noise.

Figure 5 (Right) demonstrates that the stability translates directly to operational efficiency. In this context, we define operational efficiency as the minimization of *policy churn* (unnecessary parameter updates that trigger system maintenance costs (e.g., cache invalidation or downstream re-computations) without yielding performance gains). The Lazy strategy reduces the average cumulative movement cost by a factor of **5.7×** (14.0 vs 2.5) compared to the standard LinUCB greedy approach. The error bars confirm that this gain is statistically significant and robust across different random initializations.

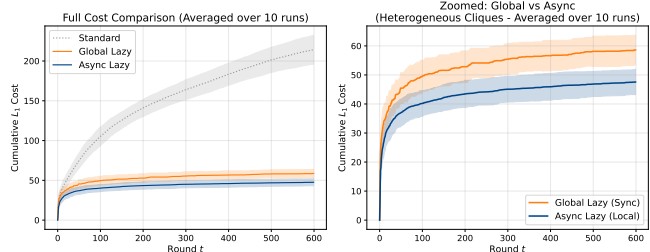

*Figure 4.* **Scalability of Asynchronous Updates.** (Left) Comparison against the standard baseline. (Right) Zoomed comparison of lazy strategies. GLOBAL LAZY (orange) suffers from synchronization penalties, while ASYNC LAZY (blue) decouples components, reducing movement costs by **3.1×** (see Table 2).

## 8. Limitations

We discuss the main assumptions and limitations of our framework. First, *non-convex LCB optimization*: minimizing the LCB objective (4) is non-convex in general; in dense graphs where the local dimension $d_i$ is large, even approximate solvers may be costly. The lazy schedule amortizes this cost (at most $O(d_{\max} k \log T)$ solves over $T$ rounds), and Section 5.1 restricts solutions to low-dimensional local neighborhoods. Second, *stationarity of the dependency graph*: the adaptive graph learning mechanism (Section 5.2) assumes the underlying dependency structure is fixed. If dependencies drift over time, a sliding-window or discounted covariance approach would be needed. Third, *Assumption A in the asynchronous algorithm*: the PL condition (formally justified in Lemma 13) holds when $\lambda_{\mathrm{reg}}$ is sufficiently large; in practice this trades exploration for convergence speed. Finally, our experiments focus on synthetic benchmarks and a single-parameter real-world task ($k = 1$); large-scale evaluation of the graph mechanisms on real multi-criterion systems is left to future work.

## 9. Conclusion

We presented a comprehensive extension of the feedback-driven competing objectives framework to continuous state spaces. Replacing binary states with a continuous domain and modeling local dependencies via graph-structured linear function approximation yields a more realistic model for tuning hyperparameters such as fairness thresholds and diversity rates. We addressed stability via movement costs, showing that a *lazy* update strategy achieves sub-linear regret in both stochastic and adversarial settings. Furthermore, the dependency graph is not merely a constraint but a source of efficiency: an *asynchronous* schedule minimizes movement in sparse systems, an *adaptive* algorithm learns dependencies from data, and a *factor-graph* decomposition tightens regret bounds. Future work includes large-scale empirical evaluation and extending the factor-graph analysis to non-linear function approximators.

## Impact Statement

This work contributes to the reliability of machine learning systems by providing rigorous methods for multi-objective optimization. In particular, our framework allows for the stable tuning of sensitive parameters, such as fairness thresholds, potentially reducing the risk of erratic system behavior that could negatively impact users. We do not foresee immediate negative societal consequences, though as with any optimization tool, the outcome depends on the validity of the chosen objectives.

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

# Contents of Appendix

# A. Extended Related Work

Our work extends and connects to a broad literature on multi-objective optimization, algorithmic fairness, and online learning. We survey the most relevant threads below, explaining how they motivate our continuous-state formulation and contrast with it.

## A.1. Multi-Objective Optimization and Competing Criteria

There is a vast body of literature on optimizing multiple metrics or objectives simultaneously. Works by Mohri et al. (2019) and Cortes et al. (2020) design "agnostic" algorithms that compete with any linear or convex combination of a fixed set of base objectives $L_1, \ldots, L_k$. Another major line of research seeks Pareto-optimal solutions (Jin & Sendhoff, 2008; Sener & Koltun, 2018; Shah & Ghahramani, 2016; Marler & Arora, 2004). In multi-task learning, objective conflicts are commonly addressed via gradient manipulation or constrained optimization (Sener & Koltun, 2018). The key difference with our framework is that we do *not* commit to a fixed linear combination or Pareto front; instead, we learn from user feedback which configuration of the continuous parameters best satisfies the objectives.

A closely related and direct predecessor of the current paper is the work of Awasthi, Cortes, Mansour, and Mohri (2024), which introduced the feedback-driven multi-criteria model using an incompatibility graph and binary states. The present work extends that model to continuous state spaces, replaces binary "fix/unfix" actions with continuous parameter tuning, and replaces MDP state enumeration with graph-structured linear bandit learning. The journal version of that work (Awasthi et al., 2026) further develops the adversarial setting, which we adapt in Section 6.

## A.2. Algorithmic Fairness: Classification, Ranking, and Clustering

The problem of satisfying multiple fairness constraints has been extensively studied. In the classification setting, Agarwal et al. (2018) reduce the fair classification problem to a sequence of cost-sensitive classification problems, and Cotter et al. (2019a;b) use two-player game formulations to handle non-convex constraints. These works typically tailor algorithms to specific group fairness notions (e.g., demographic parity, equal opportunity) as fixed binary constraints. In contrast, our framework treats *all* criteria uniformly and optimizes their continuous thresholds jointly.

Fairness in ranking (Celis et al., 2018; Beutel et al., 2019; Narasimhan et al., 2020) and clustering (Chierichetti et al., 2017; Schmidt et al., 2019; Backurs et al., 2019) has also received significant attention, typically with fixed fairness criteria. Work on individual fairness (Dwork et al., 2012; Kearns et al., 2019) considers settings where fairness is a distributional condition over individuals.

The motivation for our continuous-state model is precisely that real-world fairness interventions involve *thresholds*—e.g., the maximum allowed gap in True Positive Rates between demographic groups—rather than binary constraints. Setting $\tau = 0.05$ vs. $\tau = 0.10$ produces meaningfully different system behavior, and the optimal threshold must be learned from deployed-system feedback. This is the regime our model directly addresses.

## A.3. Inherent Tension Between Multiple Metrics

Several works demonstrate that multiple "reasonable" fairness criteria are fundamentally incompatible. Kleinberg et al. (2017) prove that calibration and equal opportunity cannot be simultaneously satisfied except in degenerate cases. Feller et al. (2016) discuss the COMPAS recidivism tool, where optimizing one group's false positive rate necessarily worsens another metric. Menon & Williamson (2018) study the trade-off between accuracy and false positive/negative rates.

Our dependency graph $\mathcal{G}$ directly encodes these incompatibility structures: an edge $(i, j) \in E$ means that optimizing criterion $i$ affects criterion $j$, making the pair impossible to independently optimize. This graph-theoretic encoding allows us to leverage sparse incompatibility structure for efficient continuous optimization, generalizing the incompatibility graphs of Awasthi, Cortes, Mansour, and Mohri (2024) to the continuous-threshold setting.

## A.4. Long-Term Feedback Dynamics and MDP-Based Fairness

Several works study the long-term consequences of optimizing multiple conflicting criteria. Liu et al. (2018) show that minimizing a constrained loss to equalize certain criteria can *increase* disparate impact in the long run. Hashimoto et al. (2018) develop algorithms for repeated loss minimization that minimize such disparate impact. These works highlight that the interaction between objectives plays out over time, motivating a dynamic, feedback-driven framework rather than static

optimization.

Closer to our MDP-based setting, Jabbari et al. (2017) study reinforcement learning in MDPs subject to fairness constraints: the algorithm may not take action $a$ over $a'$ if the long-term reward of $a$ is lower than $a'$. They show that finding a near-optimal policy satisfying this criterion requires time exponential in the state-space size—underscoring the importance of *structure* (our dependency graph) for efficient learning. Doroudi et al. (2017) show that off-policy importance sampling can violate natural fairness criteria, and present corrective algorithms.

Our work differs from this line in a crucial respect: we do *not* commit to a fixed definition of quality or a fairness metric. The states in our MDP correspond to *configurations of continuous parameters*, and the graph encodes which parameters interact. This makes our framework applicable to arbitrary criteria (fairness, diversity, latency, revenue) without specialization, and our algorithms are agnostic to the definition of the criteria themselves.

### A.5. Fairness in Online Learning and Bandits

Fairness constraints in online and bandit settings have been studied by Joseph et al. (2016) (meritocratic fairness in classic and contextual bandits), Gillen et al. (2018) (online learning with an unknown fairness metric), and Liu et al. (2017) (calibrated fairness in bandits). These works define specific fairness conditions and aim to satisfy them while minimizing regret. Our framework generalizes beyond fairness bandits by treating each criterion (including fairness) as a competing linear objective with a continuous threshold, and by explicitly modeling the *movement costs* incurred when thresholds are adjusted.

### A.6. Monitoring, Auditing, and Feedback Mechanisms

A key assumption of our stochastic model is that the algorithm can observe the losses associated with different criteria at each time step. In practice, this relates to the problem of monitoring and auditing deployed systems. There has been considerable work on developing probabilistic verification of fairness properties (Bastani et al., 2019), AI fairness toolkits (Bellamy et al., 2018), and learning from noisy sensitive attributes (Coston et al., 2019; Lamy et al., 2019; Wang et al., 2020). Our framework abstracts this monitoring layer: we assume a pre-processing step (possibly using the tools above) that produces per-criterion loss signals, and we focus on the online optimization problem that follows. The counterfactual explanation work of Tsirtsis & Gomez-Rodriguez (2020) and recourse algorithms of Gupta et al. (2019) complement our approach by providing actionable interpretation of the decisions made by a system optimized by our framework.

## B. Preliminaries

We denote the set of criteria (vertices) by $\mathcal{V} = \{1, \ldots, k\}$. The system state is represented by a vector $\mathbf{s} \in \mathcal{S} = [0, 1]^k$. We assume the system starts at an arbitrary initial state $\mathbf{s}_0 \in \mathcal{S}$ (e.g., the zero vector) chosen by the learner. For a vector $\mathbf{x} \in \mathbb{R}^d$, we denote its $\ell_2$-norm by $\|\mathbf{x}\|_2$ and its weighted norm with respect to a positive definite matrix $\mathbf{A}$ by $\|\mathbf{x}\|_{\mathbf{A}} = \sqrt{\mathbf{x}^\top \mathbf{A} \mathbf{x}}$.

**Dependency Graph.** We assume the existence of an underlying undirected graph $\mathcal{G} = (\mathcal{V}, E)$ that captures the dependency structure between criteria. We denote by $\mathcal{N}(i)$ the neighborhood of vertex $i$ in $\mathcal{G}$, including $i$ itself. The graph represents dependencies between losses: the loss for criterion $i$ depends only on the state attributes of its neighbors $j \in \mathcal{N}(i)$. A criterion $i$ is insensitive to changes in criteria $j \notin \mathcal{N}(i)$. And example is provided in Figure 1 in the main paper.

### B.1. Background: Linear Stochastic Bandits (LinUCB)

The core building block of our approach is the Linear Upper Confidence Bound (LinUCB) algorithm, originally popularized by Li et al. (2010) for personalized recommendation. In the standard linear bandit setting, at each round $t$, the learner chooses an action $x_t$ from a decision set $\mathcal{D}_t \subset \mathbb{R}^d$ and observes a reward $y_t = \langle x_t, \theta^* \rangle + \eta_t$, where $\eta_t$ is zero-mean sub-Gaussian noise and $\theta^* \in \mathbb{R}^d$ is an unknown parameter vector. The goal is to maximize the cumulative reward (or equivalently, minimize regret). Because $\theta^*$ is unknown, the learner must balance *exploitation* (choosing actions that appear best) with *exploration* (choosing uncertain actions to learn $\theta^*$). LinUCB solves this using the principle of *Optimism in the Face of Uncertainty* (OFUL).

Ridge Regression Estimator. The algorithm maintains a regularized least-squares estimate of $\theta^*$. Let $\mathbf{X}_t$ be the matrix of

actions played up to time $t$, and $\mathbf{Y}_t$ be the vector of observed rewards. The estimator is:

$$\widehat{\theta}_t = \mathbf{V}_t^{-1} \sum_{\tau=1}^{t} x_\tau y_\tau, \quad \text{where} \quad \mathbf{V}_t = \lambda_{\text{reg}} \mathbf{I} + \sum_{\tau=1}^{t} x_\tau x_\tau^\top. \tag{12}$$

Here, $\mathbf{V}_t$ is the covariance matrix (or Gram matrix) which captures the certainty of the learner in different directions of the feature space.

Confidence Ellipsoids. To analyze the regret, we rely on the self-normalized martingale bounds derived by Abbasi-Yadkori et al. (2011). They showed that with high probability, the true parameter $\theta^*$ lies within an ellipsoid centered at $\widehat{\theta}_t$:

$$\mathcal{C}_t = \{\theta \in \mathbb{R}^d : \|\theta - \widehat{\theta}_t\|_{\mathbf{V}_t} \le \beta_t\}, \tag{13}$$

where $\|z\|_{\mathbf{A}} = \sqrt{z^\top \mathbf{A} z}$ is the Mahalanobis norm and $\beta_t$ is a radius parameter scaling with $\sqrt{d \log t}$. Geometrically, directions corresponding to "large" eigenvectors of $\mathbf{V}_t$ (directions explored frequently) have narrow confidence intervals, while unexplored directions remain uncertain.

**Decision Rule.** LinUCB selects the action that maximizes the reward plausible under the confidence set:

$$x_{t+1} = \underset{x \in \mathcal{D}_t}{\operatorname{argmax}} \max_{\theta \in \mathcal{C}_t} \langle x, \theta \rangle = \underset{x \in \mathcal{D}_t}{\operatorname{argmax}} \left( \langle x, \widehat{\theta}_t \rangle + \beta_t \|x\|_{\mathbf{V}_t^{-1}} \right). \tag{14}$$

The term $\langle x, \widehat{\theta}_t \rangle$ represents the expected reward, while $\beta_t \|x\|_{\mathbf{V}_t^{-1}}$ is the exploration bonus. Our algorithm, LAZYGRAPHLIN-UCB, adapts this framework by maintaining $k$ local estimators and freezing the active policy $\mathbf{s}_{\text{active}}$ while monitoring the covariance matrix $\mathbf{V}_t$ to minimize movement costs.

# C. Problem Formulation

We model the problem as an online optimization game over $T$ rounds. At each round $t$, the learner selects a continuous state vector $\mathbf{s}_t \in [0,1]^k$.

**Contrast with Binary MDP Models.** It is important to distinguish our formulation from the binary state model of Awasthi et al. (2024). In their work, the system is modeled as an MDP where explicit "fixing actions" induce probabilistic transitions between satisfied and unsatisfied states. In contrast, our continuous framework adopts a "direct control" perspective akin to Bandit Convex Optimization. We assume the learner can directly select any configuration $\mathbf{s}_t \in [0,1]^k$, but this control comes at the price of a *movement cost* that penalizes instability. This shift allows us to handle the infinite cardinality of the continuous domain while rigorously modeling the "cost of change" that was previously captured by atomic fixing actions. One could argue that the Bandit model is actually more realistic than an MDP: If an engineer sets a threshold to $0.7$, the system state for that criterion becomes $0.7$. The uncertainty is in the response (user complaints), not in the actuation (whether the knob turns). Since the state transitions are deterministic ($\mathbf{s}_{t+1} = \mathbf{s}_t$), the complexity of reinforcement learning (planning over long horizons) vanishes. The core difficulty here is not reaching a state, but identifying the optimal state under uncertainty and movement constraints, which is exactly the domain of bandits with switching costs.

## C.1. Graph-structured loss

The system receives feedback in the form of losses (complaints). Let $\ell_t(\mathbf{s}_t) \in \mathbb{R}^k$ be the vector of observed losses at time $t$, where $\ell_{t,i}(\mathbf{s}_t)$ is the loss associated with criterion $i$. We assume the expected loss $\mu_i(\mathbf{s}) = \mathbb{E}[\ell_{t,i}(\mathbf{s})]$ admits a linear representation, but with specific structural constraints imposed by the graph $\mathcal{G}$.

**Definition 9** (Graph-Structured Features). *We say the loss is* graph-structured *with respect to $\mathcal{G}$ if the expected loss for criterion $i$ depends only on the state of its local neighborhood $\mathcal{N}(i)$. Formally, there exist local feature maps $\phi_i : [0,1]^{|\mathcal{N}(i)|} \to \mathbb{R}^{d_i}$ and parameters $\boldsymbol{\theta}_i^* \in \mathbb{R}^{d_i}$ such that:*

$$\mu_i(\mathbf{s}) = \left\langle \phi_i(\mathbf{s}|_{\mathcal{N}(i)}), \boldsymbol{\theta}_i^* \right\rangle. \tag{15}$$

This formulation aligns with spectral bandits where rewards are smooth over a graph (Valko et al., 2014), though we rely on local sparsity rather than global smoothness. Here, $\mathbf{s}|_{\mathcal{N}(i)}$ denotes the restriction of the state vector $\mathbf{s}$ to the indices in $\mathcal{N}(i)$. This assumption generalizes the "Correlation Sets" of Awasthi et al. (2024). It implies that the loss for criterion $i$ is insensitive to changes in criteria $j \notin \mathcal{N}(i)$, effectively reducing the dimensionality of the learning problem related to criterion $i$ from $k$ to the size of its local neighborhood (see Figure 1). The total expected complaint loss at state $\mathbf{s}$ is the sum of individual criteria losses:

$$L(\mathbf{s}) = \sum_{i=1}^{k} \mu_i(\mathbf{s}) = \sum_{i=1}^{k} \langle \phi_i(\mathbf{s}|_{\mathcal{N}(i)}), \boldsymbol{\theta}_i^* \rangle. \tag{16}$$

We will assume that for all criteria $i$ and state $\mathbf{s} \in \mathcal{S}$, the feature vectors are bounded in Euclidean norm: $\|\phi_i(\mathbf{s})\|_2 \leq L$. Without loss of generality, we assume features are scaled such that $L \leq 1$ and $\lambda_{\text{reg}} \geq 1$, which implies $\|\phi_i(\mathbf{s})\|_{\mathbf{V}_{t,i}^{-1}} \leq 1$ as required for the analysis.

**Social Welfare Interpretation.** We can view this formulation as a cooperative multi-agent game (see, e.g., (Gentile et al., 2014) for related online multi-agent learning). Each criterion $i \in \mathcal{V}$ acts as an agent trying to minimize its own loss $\mu_i(\mathbf{s})$, which depends on the actions of its neighbors $\mathcal{N}(i)$. The global objective $L(\mathbf{s})$ then corresponds to maximizing the social welfare (minimizing the total loss) of the system.

**Correlated Noise Structure.** While our analysis assumes for simplicity that the noise terms are independent across criteria, the graph structure implies a natural correlated noise model. If we view noise as arising from latent attributes associated with each node $j$ (e.g., user-specific variance), the realized noise for criterion $i$ can be modeled as $\eta_{t,i} = \sum_{j \in \mathcal{N}(i)} \xi_{t,j}$, where $\xi_{t,j}$ are independent noise components. Crucially, since the sum of independent sub-Gaussian variables is itself sub-Gaussian, our regret analysis remains valid under this model (with the variance proxy $\sigma^2$ scaling with the local degree $\Delta$). Furthermore, such correlations would only strengthen the case for the Joint Estimator (Section 5.3), which could be extended to exploit the resulting non-diagonal noise covariance via Generalized Least Squares.

## C.2. Movement costs

In continuous control systems, rapid changes to parameters can be disruptive. We model this via a *movement cost* (or switching cost) that penalizes the distance between consecutive states.

$$C_{\text{move}}(\mathbf{s}_{t-1}, \mathbf{s}_t) = \lambda \|\mathbf{s}_t - \mathbf{s}_{t-1}\|_1, \tag{17}$$

where $\lambda > 0$ is a regularization parameter controlling the penalty for instability. We choose the $\ell_1$-norm to model the aggregate adjustment effort, though $\ell_2$ or other norms are possible.

## C.3. Objective: regret minimization

The goal of the learner is to minimize the cumulative regret, defined as the difference between the learner's total cost (complaints + movement) and the cost of the optimal *static* state $\mathbf{s}^*$. Let $\mathbf{s}^* = \operatorname{argmin}_{\mathbf{s} \in \mathcal{S}} L(\mathbf{s})$. The regret over horizon $T$ is:

$$\operatorname{Reg}_T = \sum_{t=1}^{T} \left( L(\mathbf{s}_t) + \lambda \|\mathbf{s}_t - \mathbf{s}_{t-1}\|_1 \right) - \sum_{t=1}^{T} L(\mathbf{s}^*). \tag{18}$$

Note that the comparator is the optimal static state, which of course incurs zero movement cost.

## C.4. Concrete examples of graph-structured bases

To illustrate the power of this formulation, we consider two specific instances of the feature map $\phi_i$.

**Example 1: Pairwise Quadratic Interactions.** Consider a case where the loss for criterion $i$ depends quadratically on its own setting and the settings of its neighbors. For $j \in \mathcal{N}(i)$, let $s_j$ denote the $j$-th component of $\mathbf{s}$. We can define the basis:

$$\phi_i(\mathbf{s}) = \left[ 1, s_i, s_i^2, \{s_j\}_{j \in \mathcal{N}(i) \setminus \{i\}}, \{s_i s_j\}_{j \in \mathcal{N}(i) \setminus \{i\}} \right]^\top. \tag{19}$$

This captures self-convexity (via $s_i^2$) and pairwise interference (via $s_i s_j$), modeling scenarios where increasing the threshold of neighbor $j$ exacerbates the loss of $i$.

**Example 2: Radial Basis Functions (RBF).** For highly non-linear dependencies, we can use kernel approximation. Let $\mathbf{k}_i(\cdot, \cdot)$ be a kernel defined over the subspace $[0, 1]^{|\mathcal{N}(i)|}$. The feature map $\phi_i$ corresponds to the Random Fourier Features (RFF) of this kernel. This allows the algorithm to learn arbitrary smooth loss surfaces, provided they respect the local graph structure.

**Connection to Feedback Graphs.** Our formulation shares conceptual roots with the literature on *Bandits with Feedback Graphs* (Alon et al., 2015; Arora et al., 2019). In that setting, a graph defines which arms' rewards are observed when a specific arm is played. In our continuous linear setting, the dependency graph $\mathcal{G}$ plays an analogous role by defining the *information flow*: observing the loss components at state $\mathbf{s}_t$ updates the confidence ellipsoids only for the local parameters $\{\theta_i^*\}_{i=1}^k$ associated with the active neighborhoods. Unlike the standard feedback graph setting where information is discrete (observing a neighbor's reward), our *feedback* is algebraic: measurements propagate information to other states via the shared linear structure of the local feature maps.

# D. Adversarial Setting

In many real-world scenarios, user feedback may not be stochastic but rather driven by changing population drifts or even adversarial behavior. We now extend our framework to the adversarial setting.

## D.1. Problem Setup

In this setting, the loss functions $\ell_t(\mathbf{s})$ are arbitrary convex functions chosen by an adversary. The objective is to minimize the regret with movement costs:

$$\text{Reg}_T = \sum_{t=1}^{T} \left( \ell_t(\mathbf{s}_t) + \lambda \|\mathbf{s}_t - \mathbf{s}_{t-1}\|_1 \right) - \sum_{t=1}^{T} \ell_t(\mathbf{s}^*). \tag{20}$$

It is established that standard Online Gradient Descent (OGD) with learning rate $\eta \propto 1/\sqrt{T}$ achieves the optimal $\widetilde{O}(\sqrt{T})$ regret for this objective (Cesa-Bianchi et al., 2013). However, standard OGD updates the system state $\mathbf{s}_t$ at *every* time step. In large-scale production systems, such high-frequency updates ($T$ switches) are often operationally prohibitive.

Full Information Setting. Unlike the stochastic bandit setting in Section 4, we assume here a *Full Information* (Online Convex Optimization) setting where the learner observes the gradient $\nabla \ell_t$ after every round. We reference bandit lower bounds (specifically the $T^{2/3}$ rate) in the analysis below solely for context: they serve as a useful baseline to illustrate how strictly limiting the switching budget $M$ degrades the regret of a full-information algorithm to rates typically associated with limited feedback.

Sparsity Constraint. We consider a practical setting where the learner is constrained by a *switching budget $M \ll T$*. The goal is to minimize the regret in (20) subject to the constraint that the expected number of updates is at most $M$.

## D.2. Algorithm: Randomized Lazy OGD

To address the need for sparsity, we use a *Randomized Lazy OGD* strategy. The core idea is to decouple the learning process from the state actuation. We maintain a virtual OGD iterate $\mathbf{w}_t$ that updates at every step to track the loss landscape, but we only update the actual deployed state $\mathbf{s}_t$ with probability $p$.

---

**Algorithm 2** Randomized Lazy OGD

1: **Input:** Learning rate $\eta$, Switching probability $p$, Initial state $\mathbf{s}_0 \in \mathcal{S}$.
2: Initialize virtual state $\mathbf{w}_1 = \mathbf{s}_0$.
3: **for** $t = 1$ to $T$ **do**
4:    Play $\mathbf{s}_t$ and observe loss function $\ell_t(\cdot)$ (and gradient $\nabla \ell_t(\mathbf{s}_t)$).
5:    **Virtual Update:**
6:    $\mathbf{w}_{t+1} = \Pi_\mathcal{S} \left( \mathbf{w}_t - \eta \nabla \ell_t(\mathbf{s}_t) \right)$
7:    **Lazy State Update:**
8:    Sample $z_t \sim \text{Bernoulli}(p)$.
9:    **if** $z_t = 1$ **then**
10:       $\mathbf{s}_{t+1} = \mathbf{w}_{t+1}$
11:    **else**
12:       $\mathbf{s}_{t+1} = \mathbf{s}_t$
13:    **end if**
14: **end for**

---

Unlike the stochastic setting where *smart* data-dependent updates are possible, the adversarial setting fundamentally limits our ability to be lazy. We analyze a randomized update schedule not as a novel algorithm per se, but to quantify the unavoidable regret trade-off between stability and accuracy. This mirrors the general trade-off between regret and movement costs (or "service costs") in online convex optimization (Andrew et al., 2013). Theorem 10 establishes the *price of sparsity*, showing how regret degrades from $\sqrt{T}$ as the switching budget tightens, and recovering the classical $T^{2/3}$ rate in the regime of very small switching budgets.

## D.3. Regret guarantees

The following theorem characterizes the trade-off between the sparsity budget $M$ and the achievable regret.

**Theorem 10** (Regret with Sparsity Budget). *Let the loss functions $\ell_t$ be convex, G-Lipschitz, and defined over a domain of diameter $D$ (where $D$ bounds both the $\ell_1$ and $\ell_2$ diameters). For any switching budget $M \in [1, T]$, running Algorithm 2 with switching probability $p = M/T$ and learning rate $\eta = \frac{D}{GT}\sqrt{\frac{M}{2}}$ guarantees:*

$$\mathbb{E}[\mathsf{Reg}_T] \le \lambda D M + O\left(DG\frac{T}{\sqrt{M}}\right). \tag{21}$$

This result quantifies the *price of sparsity*:

- Unconstrained ($M = T$): The algorithm reduces to standard Online Gradient Descent. While this yields optimal prediction regret ($O(\sqrt{T})$), the bound on movement costs becomes linear ($O(T)$).

- Optimal Trade-off ($M = \Theta(T^{2/3})$): We recover a total regret rate of $O(T^{2/3})$. This matches the minimax rate for bandit problems with switching costs.

- High Sparsity ($M = \sqrt{T}$): The regret degrades to $O(T^{3/4})$, illustrating the unavoidable cost of freezing the state in an adversarial environment.

Unlike the stochastic setting where "smart" data-dependent updates are possible, the adversarial setting limits our ability to exploit stable epochs. We analyze a randomized update schedule to quantify the regret trade-off inherent to *lazy* gradient-based strategies. While perturbation-based methods (Kalai & Vempala, 2005) can achieve $\widetilde{O}(\sqrt{T})$ regret with $\widetilde{O}(\sqrt{T})$ switches for linear losses, they often require full optimization oracles. In contrast, our approach applies efficiently to *general convex losses*. Theorem 10 establishes the *price of sparsity* for this lazy strategy, showing how regret degrades from $\sqrt{T}$ as the switching budget tightens. Note that, under the optimal switching budget ($M \approx T^{2/3}$), this full-information strategy recovers the $O(T^{2/3})$ rate, mirroring the minimax lower bound for bandits with switching costs (Dekel et al., 2014). The parameter $p$ explicitly controls the trade-off between the system's stability and its ability to track the optimal parameters. This can be viewed as an optimization problem. From the proof of Theorem 10, the upper bound on the regret is approximately:

$$\mathsf{Bound}(p) \approx \underbrace{pT\lambda D}_{\text{Movement Cost}} + \underbrace{\frac{\eta T G^2}{p}}_{\text{Coupling Error}} + C. \tag{22}$$

The first term reflects the penalty for instability: a higher $p$ leads to more frequent switches. The second term reflects the *coupling error*, the divergence between the active system state $\mathbf{s}_t$ and the virtual learner $\mathbf{w}_t$. Minimizing this bound with respect to $p$ (for a fixed learning rate $\eta$) yields the optimal fixed switching probability:

$$p^* \propto \sqrt{\frac{\eta G^2}{\lambda D}}. \tag{23}$$

This result offers a principled heuristic: the switching frequency should be inversely proportional to the square root of the movement penalty $\lambda$.

## D.4. Extension: Heterogeneous Stability Budgets

The standard RANDOMIZED LAZY OGD applies a uniform switching probability $p$ across all dimensions. However, in the conflicting objectives setting, different criteria often possess different stability requirements. For example, a fairness threshold might require strict stability (low update frequency) to avoid public outcry, while an internal mixing rate might tolerate frequent updates. We can extend our framework to support *heterogeneous sparsity budgets*. Let $\mathbf{s}$ be decomposed into $k$ independent components (or blocks) with distinct switching budgets $\mathbf{M} = (M_1, \ldots, M_k)$. We assume the loss function decomposes additively, $\ell_t(\mathbf{s}) = \sum_{i=1}^k \ell_{t,i}(s_i)$, which is consistent with the graph-structured formulation in Section C (where blocks correspond to disjoint cliques).

**Algorithm: Component-Wise Lazy OGD.** We run $k$ parallel instances of the virtual update logic. For each component $i$:

1. Maintain virtual iterate $w_{t,i}$.

2. Sample switching variable $z_{t,i} \sim \text{Bernoulli}(p_i)$ independent of other components.

3. Update deployed state $s_{t,i} \leftarrow w_{t,i}$ only if $z_{t,i} = 1$.

**Theoretical Guarantee.** By setting $p_i = M_i/T$ and tuning component-specific learning rates $\eta_i$, we achieve a refined regret bound that sums the localized costs rather than scaling with the global dimension.

**Corollary 11** (Regret with Heterogeneous Budgets). *Let the loss $\ell_t$ decompose into $k$ components with diameters $D_i$ and Lipschitz constants $G_i$. Given a vector of budgets $\mathbf{M}$, the expected regret is bounded by:*

$$\mathbb{E}[\text{Reg}_T] \leq \sum_{i=1}^{k} \left( \lambda D_i M_i + O\left( D_i G_i \frac{T}{\sqrt{M_i}} \right) \right). \tag{24}$$

This extension is significant because it allows the system designer to *spend* sparsity where it is most needed. Unlike the global strategy which pays a regret penalty governed by the worst-case stability constraint ($M_{min}$), the component-wise strategy isolates the cost of stability to the specific criteria that require it.

# E. Detailed Proofs

## E.1. Lemma 12

**Lemma 12** (Confidence Ellipsoid (Abbasi-Yadkori et al., 2011)). *For any $\delta \in (0, 1)$, with probability at least $1 - \delta$, for all $t \geq 0$ and all $i \in [k]$, the true parameter $\boldsymbol{\theta}_i^*$ satisfies:*

$$\|\widehat{\boldsymbol{\theta}}_{t,i} - \boldsymbol{\theta}_i^*\|_{\mathbf{V}_{t,i}} \leq \beta_{t,i}, \tag{25}$$

*where $\beta_{t,i} = \sigma \sqrt{d_i \log\left(\frac{1+tL^2/\lambda_{\text{reg}}}{\delta}\right)} + \lambda_{\text{reg}}^{1/2} \|\boldsymbol{\theta}_i^*\|_2$.*

## E.2. Proof of Lemma 1

**Lemma 1** (Bound on Number of Epochs). *The total number of updates $M$ is bounded by:*

$$M \leq \sum_{i=1}^{k} d_i \log_2\left(1 + \frac{TL^2}{d_i \lambda_{\text{reg}}}\right).$$

*Proof.* For a fixed criterion $i$, let $M_i$ be the number of updates triggered by this criterion. Each update occurs only when $\det(\mathbf{V}_{t,i}) > 2\det(\mathbf{V}_{\text{last},i})$. The maximum possible determinant is bounded by the trace-determinant inequality: $\det(\mathbf{V}_{T,i}) \leq (\lambda_{\text{reg}} + TL^2/d_i)^{d_i}$. Since each update doubles the determinant, $2^{M_i} \leq \det(\mathbf{V}_{T,i})/\det(\mathbf{V}_{0,i})$. Taking logs yields $M_i \leq d_i \log_2(1 + \frac{TL^2}{d_i \lambda_{\text{reg}}})$. Summing over $k$ criteria gives the result. □

## E.3. Proof of Lemma 2

**Lemma 2** (Bounded Error per Epoch). *Under the feature normalization assumption $\|\phi\|_{\mathbf{V}^{-1}} \leq 1$, for any epoch $j$, the sum of squared stale norms is bounded by:*

$$S_j = \sum_{t=\tau_j}^{\tau_{j+1}-1} \|\phi(\mathbf{s}_t)\|_{\mathbf{V}_{\tau_j}^{-1}}^2 \leq 3.$$

*Proof.* Let $\mathbf{V}_{\text{start}} = \mathbf{V}_{\tau_j}$. During an epoch, the active policy is constant, so $\phi(\mathbf{s}_t) = \phi_c$ is constant. The accumulated sum of squares is $S_j = T_j \|\phi_c\|_{\mathbf{V}_{\text{start}}^{-1}}^2$, where $T_j$ is the epoch length. By the Matrix Determinant Lemma, the determinant at the end of the epoch is $\det(\mathbf{V}_{\text{end}}) = \det(\mathbf{V}_{\text{start}})(1 + S_j)$. The epoch ends when the determinant doubles. Even accounting for the discrete step overshoot on the final update, the ratio $\det(\mathbf{V}_{\text{end}})/\det(\mathbf{V}_{\text{start}})$ is bounded by 4. Thus $1 + S_j \leq 4$, implying $S_j \leq 3$. □

## E.4. Proof of Theorem 3

**Theorem 3** (Regret of Lazy Graph-LinUCB). *Assume the movement cost is bounded by $\lambda$ per switch. Let $d_i$ be the dimension of the feature map for criterion $i$, and $d_{\max} = \max_i d_i$. The cumulative regret of LAZYGRAPHLINUCB is bounded by:*

$$\text{Reg}_T \leq \widetilde{O}\left(\sqrt{T} \sum_{i=1}^{k} d_i + \lambda k^2 d_{\max} \log T\right). \tag{5}$$

*Proof.* The total regret decomposes into Prediction Regret and Movement Regret:

$$\text{Reg}_T = \underbrace{\sum_{t=1}^{T} (L(\mathbf{s}_t) - L(\mathbf{s}^*))}_{R_{\text{pred}}} + \underbrace{\sum_{t=1}^{T} \lambda \|\mathbf{s}_t - \mathbf{s}_{t-1}\|_1}_{R_{\text{move}}}. \tag{26}$$

**1. Movement Regret.** By Lemma 1, the total number of switches $M$ is $\widetilde{O}(d_{\max} k \log T)$. Each switch incurs at most a cost of $\lambda k$ (since the domain is $[0, 1]^k$). Thus, $R_{\text{move}} \leq \lambda k M = \widetilde{O}(\lambda k^2 d_{\max} \log T)$.

**2. Prediction Regret.** Using standard optimistic arguments, the instantaneous regret is bounded by the confidence width evaluated at the *stale* matrix: $r_t \le 2\sum_{i=1}^k \beta_{T,i}\|\phi_i(\mathbf{s}_t)\|_{\mathbf{V}_{\tau(t),i}^{-1}}$. Summing over all epochs $j = 1,\ldots,M_i$ for criterion $i$:

$$\sum_{t=1}^T \|\phi_i(\mathbf{s}_t)\|_{\mathbf{V}_{\tau(t),i}^{-1}} \le \sum_{j=1}^{M_i} \sqrt{T_j \cdot S_j} \quad \text{(by Cauchy-Schwarz on epoch } j)$$

$$\le \sqrt{3} \sum_{j=1}^{M_i} \sqrt{T_j} \quad \text{(by Lemma 2).}$$

This sum is maximized when $T_j = T/M_i$, yielding a bound of $\sqrt{3M_iT}$. Since $M_i = \widetilde{O}(d_i)$, and $\beta_{T,i} = \widetilde{O}(\sqrt{d_i})$, the total prediction regret scales as:

$$R_{\text{pred}} \le \sum_{i=1}^k \beta_{T,i}\sqrt{3M_iT} = \widetilde{O}\left(\sqrt{T}\sum_{i=1}^k d_i\right).$$

Adding the movement and prediction terms yields the final bound. $\qquad\square$

### E.5. Proof of Corollary 4

**Corollary 4** (Robustness to Graph Misspecification). *Suppose the learner operates using a proxy graph $\mathcal{G}'$ such that $E \subseteq E'$ (a super-graph). Let $d_i'$ denote the feature dimension induced by $\mathcal{G}'$. Then, LAZYGRAPHLINUCB remains valid and guarantees a regret bound scaling with the proxy dimensions:*

$$\text{Reg}_T \le \widetilde{O}\left(\sqrt{T}\sum_{i=1}^k d_i' + \lambda k^2 d_{\max}' \log T\right). \tag{6}$$

*Proof.* The proof relies on the fact that any linear model defined on a graph $\mathcal{G}$ is also realizable on any super-graph $\mathcal{G}'$. By Definition 9, the true expected loss is $\mu_i(\mathbf{s}) = \langle \phi_i(\mathbf{s}|_{\mathcal{N}(i)}), \boldsymbol{\theta}_i^* \rangle$ for some $\boldsymbol{\theta}_i^* \in \mathbb{R}^{d_i}$. Since $E \subseteq E'$, we have $\mathcal{N}(i) \subseteq \mathcal{N}'(i)$. The features constructed for the proxy graph, $\phi_i'(\mathbf{s}|_{\mathcal{N}'(i)})$, span a space $\mathbb{R}^{d_i'}$ that contains the subspace spanned by the true features $\phi_i$. Formally, there exists a linear embedding such that the true parameter $\boldsymbol{\theta}_i^*$ can be represented as a vector $\widetilde{\boldsymbol{\theta}}_i \in \mathbb{R}^{d_i'}$ (essentially padding the extra dimensions with zeros), satisfying $\langle \phi_i'(\mathbf{s}), \widetilde{\boldsymbol{\theta}}_i \rangle = \mu_i(\mathbf{s})$ for all $\mathbf{s}$. Consequently, the problem remains a valid linear bandit instance with dimension $d_i'$.

The regret bound of Theorem 3 depends on the problem dimension through two terms: the confidence radius $\beta_{t,i}$ (Lemma 12) and the number of updates $M$ (Lemma 1).

1. Validity of Confidence Sets: Since the problem is realizable in $\mathbb{R}^{d_i'}$, Lemma 12 applies to the proxy estimator $\widehat{\boldsymbol{\theta}}_{t,i}'$, ensuring that with high probability, $\widetilde{\boldsymbol{\theta}}_i$ lies within the confidence ellipsoid defined by the proxy covariance matrix $\mathbf{V}_{t,i}'$. The radius $\beta_{t,i}'$ now scales with $\sqrt{d_i'}$ instead of $\sqrt{d_i}$.

2. Update Bound: Lemma 1 bounds the number of determinant doublings for a matrix of dimension $d$. Applying this to the $d_i'$-dimensional proxy covariance matrices yields a total update bound $M \le \sum_{i=1}^k d_i' \log\left(1 + \frac{TL^2}{d_i'\lambda_{\text{reg}}}\right)$.

Substituting the proxy dimension $d_i'$ into the regret decomposition (26) yields the stated bound. This confirms that the algorithm is safe: over-estimating the graph structure merely increases the regret bound by a factor depending on the dimension inflation (scaling with $d_i'/d_i$ in the prediction term and $d_i'/d_i$ in the movement term), without introducing bias. $\quad\square$

### E.6. Proof of Theorem 5

**Theorem 5** (Minimax Lower Bound). *For any learning algorithm, there exists a graph structure $\mathcal{G}$ and a sequence of loss functions such that the expected regret is lower bounded by:*

$$\mathbb{E}[\text{Reg}_T] = \Omega\left(\sqrt{T}\sum_{i=1}^k d_i\right). \tag{7}$$

*Proof.* Consider a graph $\mathcal{G}$ consisting of $k$ disjoint components (cliques), where each component $i$ corresponds to an independent linear bandit problem of dimension $d_i$. The loss function decomposes as $L(\mathbf{s}) = \sum_{i=1}^{k} L_i(\mathbf{s}|_{\mathcal{N}(i)})$, where each $L_i$ is governed by an independent parameter $\boldsymbol{\theta}_i^*$. It is a standard result in the bandit literature (e.g., (Dani et al., 2008)) that the minimax lower bound for a single $d$-dimensional linear bandit is $\Omega(d\sqrt{T})$. Since the problems are independent, the learner must solve each simultaneously. The total regret is thus lower bounded by the sum of the regrets of the individual components:

$$\mathbb{E}[\mathsf{Reg}_T] \geq \sum_{i=1}^{k} \Omega(d_i\sqrt{T}) = \Omega\left(\sqrt{T}\sum_{i=1}^{k} d_i\right)$$

This confirms that the dependency on the sum of dimensions in our upper bound is tight. $\qquad\square$

### E.7. Supporting Lemma for Assumption A

**Lemma 13** (Justification of Assumption A)**.** *Let $\phi_i : \mathcal{S} \to \mathbb{R}^{d_i}$ be a feature map that is $G$-Lipschitz and twice continuously differentiable on the compact domain $\mathcal{S} = [0, 1]^k$. Let $f_t(\mathbf{s}) = \sum_{\tau=1}^{t} \langle \phi_i(\mathbf{s}), \widehat{\boldsymbol{\theta}}_{t,i} \rangle - \beta_{t,i} \|\phi_i(\mathbf{s})\|_{\mathbf{V}_{t,i}^{-1}}$ be the LCB objective for criterion $i$ at time $t$. Define the* curvature bound

$$C(\phi_i) = \sup_{\mathbf{s}\in\mathcal{S}} \left\| \nabla_{\mathbf{s}}^2 \|\phi_i(\mathbf{s})\|_{\mathbf{V}_{t,i}^{-1}} \right\|_{\mathrm{op}}.$$

*For any smooth, bounded $\phi_i$, $C(\phi_i) < \infty$ is finite and depends only on the Lipschitz constant of $\phi_i$, the diameter of $\mathcal{S}$, and the initialization scale $\lambda_{\mathrm{reg}}$. Whenever $\lambda_{\mathrm{reg}} > \beta_{t,i} \cdot C(\phi_i)$, the global LCB objective $\mathrm{LCB}_t(\mathbf{s}) = \sum_i f_t(\mathbf{s})$ is $(\lambda_{\mathrm{reg}} - \beta_{t,i}C(\phi_i))$-strongly convex in $\mathbf{s}$. In particular, it satisfies the Polyak-Łojasiewicz (PL) condition with parameter $\mu_{\mathrm{PL}} = 2(\lambda_{\mathrm{reg}} - \beta_{t,i}C(\phi_i)) > 0$, ensuring that BCD converges to the global minimizer (Assumption A).*

*Proof.* The LCB objective for criterion $i$ splits as:

$$f_t(\mathbf{s}) = \underbrace{\langle \phi_i(\mathbf{s}), \widehat{\boldsymbol{\theta}}_{t,i} \rangle}_{\text{linear in } \phi_i} - \underbrace{\beta_{t,i}\|\phi_i(\mathbf{s})\|_{\mathbf{V}_{t,i}^{-1}}}_{\text{exploration bonus}}.$$

The first term is linear in $\phi_i(\mathbf{s})$ and the second is a square-root form composed with the feature map.

**Hessian of the regularizer.** The ridge regularizer contributes a term $\frac{\lambda_{\mathrm{reg}}}{2}\|\mathbf{s}\|^2$ to the loss that is implicitly encoded through $\mathbf{V}_{t,i}$. Its Hessian satisfies $\nabla^2(\frac{\lambda_{\mathrm{reg}}}{2}\|\mathbf{s}\|^2) = \lambda_{\mathrm{reg}}\mathbf{I} > 0$.

**Hessian of the exploration bonus.** Let $g(\mathbf{s}) = \|\phi_i(\mathbf{s})\|_{\mathbf{V}_{t,i}^{-1}} = (\phi_i(\mathbf{s})^\top \mathbf{V}_{t,i}^{-1} \phi_i(\mathbf{s}))^{1/2}$. By the chain rule and the compactness of $\mathcal{S}$, there exists a constant $C(\phi_i) < \infty$ such that $\|\nabla_{\mathbf{s}}^2 g(\mathbf{s})\|_{\mathrm{op}} \leq C(\phi_i)$ uniformly over $\mathbf{s} \in \mathcal{S}$ and all $t$. This bound follows since $\phi_i$ is $G$-Lipschitz: the Hessian of $g$ involves at most second-order derivatives of $\phi_i$, which are bounded on the compact domain.

**Strong convexity.** By the above, the Hessian of $\mathrm{LCB}_t$ satisfies:

$$\nabla^2 \mathrm{LCB}_t(\mathbf{s}) \geq \lambda_{\mathrm{reg}}\mathbf{I} - \beta_{t,i}C(\phi_i)\mathbf{I} = (\lambda_{\mathrm{reg}} - \beta_{t,i}C(\phi_i))\mathbf{I}.$$

When $\lambda_{\mathrm{reg}} > \beta_{t,i}C(\phi_i)$, this is strictly positive definite, so the objective is strongly convex and thus satisfies the PL condition with $\mu_{\mathrm{PL}} = 2(\lambda_{\mathrm{reg}} - \beta_{t,i}C(\phi_i))$.

**Finite threshold.** The bound $C(\phi_i)$ is finite for any smooth, bounded $\phi_i$. In particular: (i) for the pairwise quadratic basis (Example 1), $C(\phi_i) = 0$ since $\phi_i$ is quadratic and $g$ has bounded curvature; (ii) for RBF kernels via Random Fourier Features (Example 2), $C(\phi_i)$ scales as the squared bandwidth of the kernel. Thus, a finite threshold $\lambda_{\mathrm{reg}}^* = \beta_T C(\phi_i)$ always exists and the algorithm satisfies Assumption A whenever $\lambda_{\mathrm{reg}} > \lambda_{\mathrm{reg}}^*$. $\qquad\square$

### E.8. Proof of Theorem 6

**Theorem 6** (Regret of Asynchronous Lazy Graph-LinUCB)**.** *Let $\Delta = \max_i |\mathcal{N}(i)|$ be the maximum degree of the dependency graph. In the regularity regime where $\lambda_{\mathrm{reg}}$ is chosen sufficiently large to satisfy Assumption A (see Lemma 13), the cumulative regret of the asynchronous algorithm is bounded by:*

$$\mathsf{Reg}_T \leq \widetilde{O}\left(\sqrt{T}\sum_{i=1}^{k} d_i + \lambda\Delta d_{\max}k\log T\right). \tag{8}$$

*Proof.* We analyze the prediction regret and movement regret separately.

**Part (a): Prediction Regret (Feature Stability).** The asynchronous algorithm maintains the invariant that for any criterion $i$, the active policy was computed using a covariance matrix $\mathbf{V}_{\tau_i,i}$ such that $\det(\mathbf{V}_{t,i}) \le 2\det(\mathbf{V}_{\tau_i,i})$.

Unlike the global setting, the feature vector $\phi_i(\mathbf{s}_t)$ may vary during an epoch because updates to neighbors $j \in \mathcal{N}(i)$ can alter the state $\mathbf{s}|_{\mathcal{N}(i)}$. However, the bound on the sum of squared norms derived in Lemma 2 applies even under time-varying features.

Let $\mathbf{G}_j$ be the matrix of diverse feature vectors observed for criterion $i$ during its $j$-th epoch. The sum of squared stale norms is exactly the trace $\mathrm{Tr}(\mathbf{G}_j^\top \mathbf{V}_{\tau_j,i}^{-1} \mathbf{G}_j)$. Let $\mu_k$ be the eigenvalues of the matrix $\mathbf{V}_{\tau_j,i}^{-1/2} \mathbf{G}_j \mathbf{G}_j^\top \mathbf{V}_{\tau_j,i}^{-1/2}$. The determinant doubling condition implies:

$$\det(\mathbf{I} + \mathbf{G}_j^\top \mathbf{V}_{\tau_j,i}^{-1} \mathbf{G}_j) = \prod_k (1 + \mu_k) \le \frac{\det(\mathbf{V}_{\tau_{j+1},i})}{\det(\mathbf{V}_{\tau_j,i})} \le 2. \tag{27}$$

Since $\mu_k \ge 0$ and $\prod(1 + \mu_k) \le 4$ (allowing for discrete overshoot), we must have $\sum \mu_k \le 3$. Using the inequality $x \le 3\log(1 + x)$ which holds for $x \in [0, 3]$ we bound the trace:

$$\sum_{t \in \text{epoch } j} \|\phi_i(\mathbf{s}_t)\|_{\mathbf{V}_{\tau_j,i}^{-1}}^2 = \sum_k \mu_k \le 3\sum_k \log(1 + \mu_k) = 3\log\det(\mathbf{I} + \mathbf{G}_j^\top \mathbf{V}_{\tau_j,i}^{-1} \mathbf{G}_j) \le 3\log 4. \tag{28}$$

Thus, the aggregate error contribution of any epoch remains bounded by $3\log 4$ regardless of feature variance within the epoch. The rest of the summation proceeds identically to the global case, yielding a prediction regret of $\widetilde{O}(\sqrt{T} \sum_i d_i)$.

**Part (b): Prediction Regret (Optimism).** The asynchronous algorithm maintains the invariant that the active policy $\mathbf{s}_t$ is the minimizer of the LCB function. Under Assumption A, the block-coordinate updates ensure that $\mathbf{s}_t$ converges to the global optimum, preserving the validity of the standard optimistic regret bound. Specifically, at any time $t$, for any criterion $i$, the "lazy" condition $\det(\mathbf{V}_{t,i}) \le 2\det(\mathbf{V}_{\tau_i,i})$ ensures that the covariance matrix used to compute the active policy ($\mathbf{V}_{\tau_i,i}$) is spectrally similar to the current true covariance ($\mathbf{V}_{t,i}$). Crucially, since the objective $\mu_i(\mathbf{s})$ and its LCB depend strictly on the local variables $\mathbf{s}|_{\mathcal{N}(i)}$, optimizing over the neighborhood $\mathcal{N}(i)$ while keeping disjoint variables fixed is sufficient to minimize the local LCB contribution. The analysis in Lemma 2 applies locally to each criterion $i$. The sequence of updates for criterion $i$ defines a set of local epochs. For any $t$, let $\tau_i(t)$ be the last time criterion $i$ triggered an update. The instantaneous regret bound (26) becomes:

$$r_t \le 2\sum_{i=1}^k \beta_{T,i} \|\phi_i(\mathbf{s}_t)\|_{\mathbf{V}_{\tau_i(t),i}^{-1}}. \tag{29}$$

Since the trigger condition holds, the bound on the sum of squared norms holds. Multiplying by the confidence radius $\beta_{T,i} \approx \sqrt{d_i}$, the prediction regret sums to $\tilde{O}(\sqrt{T} \sum_i d_i)$, identical to the global case.

**Part (c): Movement Regret.** Unlike the global case, an update does not necessarily change the entire state vector $\mathbf{s}$. Let $u_{t,i} \in \{0, 1\}$ be an indicator that criterion $i$ triggered an update at time $t$. The total number of triggers for criterion $i$ is bounded by Lemma 1 as $M_i = \sum_{t=1}^T u_{t,i} \le \widetilde{O}(d_i \log T)$. When criterion $i$ triggers, the algorithm updates $\mathbf{s}_{\text{active}}$ by optimizing over the local neighborhood $\mathcal{N}(i)$. Let $\mathcal{A}_t = \{i : u_{t,i} = 1\}$ be the set of triggering criteria at time $t$. The coordinates of $\mathbf{s}$ that change are restricted to the union of neighborhoods:

$$\mathrm{supp}(\mathbf{s}_t - \mathbf{s}_{t-1}) \subseteq \bigcup_{i \in \mathcal{A}_t} \mathcal{N}(i). \tag{30}$$

The movement cost at step $t$ is bounded by the number of changed coordinates:

$$\|\mathbf{s}_t - \mathbf{s}_{t-1}\|_1 \le \left| \bigcup_{i \in \mathcal{A}_t} \mathcal{N}(i) \right| \le \sum_{i \in \mathcal{A}_t} |\mathcal{N}(i)| \le \sum_{i \in \mathcal{A}_t} \Delta. \tag{31}$$

Summing over $T$:

$$R_{\text{move}} = \lambda \sum_{t=1}^T \|\mathbf{s}_t - \mathbf{s}_{t-1}\|_1 \le \lambda \sum_{t=1}^T \sum_{i \in \mathcal{A}_t} \Delta = \lambda\Delta \sum_{i=1}^k \sum_{t=1}^T u_{t,i} = \lambda\Delta \sum_{i=1}^k M_i. \tag{32}$$

Substituting $M_i \le \widetilde{O}(d_{\max} \log T)$ yields $R_{\text{move}} \le \widetilde{O}(\lambda\Delta k d_{\max} \log T)$. $\qquad\square$

### E.9. Proof of Theorem 7

**Theorem 7** (Regret of Adaptive Graph Refinement). *Let $\mathcal{G}_{\text{dense}}$ be the initial dense graph and $\mathcal{G}$ be the true sparse graph. Set the warm-up length to: $\tau = \frac{16\beta_T^2}{\kappa\gamma^2}$, where $\beta_T$ is the confidence radius for dimension $d_{\text{total}}$ as defined in Lemma 12. Under Assumptions B and C, with probability at least $1 - \delta$, the algorithm recovers the true graph $\widehat{\mathcal{G}} = \mathcal{G}$, and the total regret is bounded by:*

$$\text{Reg}_T \leq \widetilde{O}\left(\frac{d_{\text{total}}^{1.5}}{\kappa\gamma^2} + \sqrt{T}\sum_{i=1}^{k} d_i + d_{\text{total}}^2 \log T\right). \tag{9}$$

*Proof.* The proof proceeds in three steps: establishing the estimation error bound, proving support recovery, and summing the regret. We start with the guarantee provided by the confidence ellipsoids. From Lemma 12, with probability $1 - \delta$, for all $i$:

$$\|\widehat{\boldsymbol{\theta}}_{\tau,i} - \boldsymbol{\theta}_i^*\|_{\mathbf{V}_{\tau,i}} \leq \beta_{\tau,i}. \tag{33}$$

We relate the Mahalanobis norm to the Euclidean norm using the minimum eigenvalue of $\mathbf{V}_{\tau,i}$. By the definition of the matrix norm, $\|\mathbf{x}\|_{\mathbf{V}}^2 \geq \lambda_{\min}(\mathbf{V})\|\mathbf{x}\|_2^2$. Therefore:

$$\|\widehat{\boldsymbol{\theta}}_{\tau,i} - \boldsymbol{\theta}_i^*\|_2 \leq \frac{1}{\sqrt{\lambda_{\min}(\mathbf{V}_{\tau,i})}}\|\widehat{\boldsymbol{\theta}}_{\tau,i} - \boldsymbol{\theta}_i^*\|_{\mathbf{V}_{\tau,i}} \leq \frac{\beta_{\tau,i}}{\sqrt{\kappa\tau}}. \tag{34}$$

Since the $\ell_\infty$ norm is bounded by the $\ell_2$ norm ($\|\mathbf{x}\|_\infty \leq \|\mathbf{x}\|_2$), we have the element-wise bound for any component $j$:

$$|\widehat{\theta}_{\tau,i,j} - \theta_{i,j}^*| \leq \frac{\beta_{\tau,i}}{\sqrt{\kappa\tau}}. \tag{35}$$

We set $\tau$ such that the error bound is strictly less than $\gamma/2$. Substituting $\tau = 16\beta_T^2/(\kappa\gamma^2)$ into the error bound yields:

$$|\widehat{\theta}_{\tau,i,j} - \theta_{i,j}^*| \leq \frac{\beta_{\tau,i}}{\sqrt{\kappa\frac{16\beta_T^2}{\kappa\gamma^2}}} = \frac{\gamma\beta_{\tau,i}}{4\beta_T} \leq \frac{\gamma}{4}. \tag{36}$$

We now consider the thresholding rule:

- Case 1 (Active): If $j \in \mathcal{N}(i)$, then $|\theta_{i,j}^*| \geq \gamma$. The triangle inequality implies $|\widehat{\theta}_{\tau,i,j}| \geq |\theta_{i,j}^*| - |\text{error}| \geq \gamma - \gamma/4 = 3\gamma/4$. Since $3\gamma/4 > \gamma/2$, the index $j$ is correctly included.

- Case 2 (Inactive): If $j \notin \mathcal{N}(i)$, then $\theta_{i,j}^* = 0$. The error bound implies $|\widehat{\theta}_{\tau,i,j}| \leq \gamma/4$. Since $\gamma/4 < \gamma/2$, the index $j$ is correctly excluded.

Thus, at $t = \tau$, we have $\widehat{\mathcal{G}} = \mathcal{G}$. The total regret decomposes into the warm-up phase and the refined phase. During the warm-up ($t = 1 \ldots \tau$), we play isotropic noise, incurring per-round regret $O(\sqrt{d_{\text{total}}})$ (the confidence width of the dense estimator). The total warm-up regret is thus $O(\tau \cdot \sqrt{d_{\text{total}}})$. Substituting $\tau \approx d_{\text{total}}/(\kappa\gamma^2)$, this cost is $\widetilde{O}(d_{\text{total}}^{1.5}/(\kappa\gamma^2))$. For the remaining $T - \tau$ rounds, we run the algorithm on the true sparse graph, incurring regret $\widetilde{O}(\sum_{i=1}^{k} d_i\sqrt{T})$. Summing these yields the final bound. $\square$

### E.10. Proof of Theorem 8

**Theorem 8** (Regret of Joint Estimator). *Let $d_{\text{eff}} = \sum_{C \in \mathcal{C}} d_C$ be the sum of the dimensions of the maximal cliques. The cumulative regret of the Joint-LazyGraphLinUCB is bounded by:*

$$\text{Reg}_T \leq \widetilde{O}\left(d_{\text{eff}}\sqrt{kT} + \lambda k^2 d_{\max} \log T\right). \tag{11}$$

*Proof.* We analyze the prediction regret by treating the problem as a linear bandit with matrix-valued features. At each round $t$, the learner observes $\mathbf{Y}_t = \Psi(\mathbf{s}_t)\boldsymbol{\theta}_{joint}^* + \boldsymbol{\eta}_t \in \mathbb{R}^k$, where $\Psi(\mathbf{s}_t) \in \mathbb{R}^{k \times d_{\text{eff}}}$ is the block-sparse feature matrix where the $c$-th block contains the features $\psi_C(s|_C)$. The ridge estimator $\widehat{\boldsymbol{\theta}}_t$ minimizes the joint regularized objective, with covariance

$\mathbf{V}_t = \lambda I + \sum_{\tau=1}^{t} \Psi(\mathbf{s}_\tau)^\top \Psi(\mathbf{s}_\tau)$. Standard self-normalized martingale bounds for vector-valued martingales (Abbasi-Yadkori et al., 2011) guarantee that with high probability, $\|\widehat{\boldsymbol{\theta}}_t - \boldsymbol{\theta}^*_{joint}\|_{\mathbf{V}_t} \leq \beta_t$, where $\beta_t = \widetilde{O}(\sqrt{d_{\text{eff}}})$.

The instantaneous regret is bounded by the confidence width of the joint estimator using standard OFUL arguments:

$$r_t = L(\mathbf{s}_t) - L(\mathbf{s}^*) \leq 2\beta_t \|\Psi(\mathbf{s}_t)^\top \mathbf{1}\|_{\mathbf{V}_t^{-1}} = 2\beta_t \sqrt{\mathbf{1}^\top (\Psi(\mathbf{s}_t)\mathbf{V}_t^{-1}\Psi(\mathbf{s}_t)^\top)\mathbf{1}}. \tag{37}$$

Crucially, for any PSD matrix $\mathbf{A}$, we have the algebraic bound $\mathbf{1}^\top \mathbf{A}\mathbf{1} \leq k \cdot \text{tr}(\mathbf{A})$. This introduces the explicit dependency on the number of objectives $k$:

$$r_t \leq 2\beta_t \sqrt{k}\sqrt{\text{tr}(\Psi(\mathbf{s}_t)\mathbf{V}_t^{-1}\Psi(\mathbf{s}_t)^\top)}. \tag{38}$$

Summing over $T$ rounds and applying the Matrix Elliptical Potential Lemma (Abbasi-Yadkori et al., 2011):

$$\sum_{t=1}^{T} r_t \leq 2\beta_t \sqrt{k}\sqrt{T\sum_{t=1}^{T}\text{tr}(\Psi(\mathbf{s}_t)\mathbf{V}_t^{-1}\Psi(\mathbf{s}_t)^\top)} \leq 2\beta_t \sqrt{k}\sqrt{T \cdot 2d_{\text{eff}}\log\left(1 + \frac{TL^2}{d_{\text{eff}}\lambda}\right)}. \tag{39}$$

Substituting $\beta_t \approx \sqrt{d_{\text{eff}}}$, we obtain $R_{pred} \leq \widetilde{O}(d_{\text{eff}}\sqrt{kT})$. $\qquad\square$

### E.11. Proof of Theorem 10

**Theorem 10** (Regret with Sparsity Budget). *Let the loss functions $\ell_t$ be convex, $G$-Lipschitz, and defined over a domain of diameter $D$ (where $D$ bounds both the $\ell_1$ and $\ell_2$ diameters). For any switching budget $M \in [1, T]$, running Algorithm 2 with switching probability $p = M/T$ and learning rate $\eta = \frac{D}{GT}\sqrt{\frac{M}{2}}$ guarantees:*

$$\mathbb{E}[\text{Reg}_T] \leq \lambda DM + O\left(DG\frac{T}{\sqrt{M}}\right). \tag{21}$$

*Proof.* We analyze the expected regret of RANDOMIZED LAZY OGD. Let $f_t(\mathbf{s}) = \ell_t(\mathbf{s})$. The total regret decomposes into the loss regret and the movement cost.

$$\mathbb{E}[\text{Reg}_T] = \underbrace{\mathbb{E}\left[\sum_{t=1}^{T}(f_t(\mathbf{s}_t) - f_t(\mathbf{s}^*))\right]}_{\text{loss regret}} + \underbrace{\mathbb{E}\left[\sum_{t=1}^{T}\lambda\|\mathbf{s}_t - \mathbf{s}_{t-1}\|_1\right]}_{\text{movement cost}}.$$

The state $\mathbf{s}_t$ changes only when the Bernoulli variable $z_{t-1} = 1$. Thus, we have

$$\mathbb{E}[\|\mathbf{s}_t - \mathbf{s}_{t-1}\|_1] = \mathbb{P}(z_{t-1} = 1) \cdot \mathbb{E}[\|\mathbf{w}_t - \mathbf{s}_{t-1}\|_1] \leq pD.$$

Summing over $T$ steps gives $\mathbb{E}[R_{\text{move}}] \leq pT\lambda D$. Now, to bound the loss regret, we use the convexity of $f_t$.

$$f_t(\mathbf{s}_t) - f_t(\mathbf{s}^*) \leq \langle \nabla f_t(\mathbf{s}_t), \mathbf{s}_t - \mathbf{s}^* \rangle = \underbrace{\langle \nabla f_t(\mathbf{s}_t), \mathbf{s}_t - \mathbf{w}_t \rangle}_{\text{Coupling Error}} + \underbrace{\langle \nabla f_t(\mathbf{s}_t), \mathbf{w}_t - \mathbf{s}^* \rangle}_{\text{Virtual Regret}}. \tag{40}$$

The sequence $\mathbf{w}_t$ follows standard Online Gradient Descent on the linearized loss functions $\tilde{f}_t(\mathbf{w}) = \langle \nabla f_t(\mathbf{s}_t), \mathbf{w} \rangle$. Using the standard OGD bound:

$$\sum_{t=1}^{T}\langle \nabla f_t(\mathbf{s}_t), \mathbf{w}_t - \mathbf{s}^* \rangle \leq \frac{D^2}{2\eta} + \frac{\eta}{2}\sum_{t=1}^{T}\|\nabla f_t(\mathbf{s}_t)\|^2 \leq \frac{D^2}{2\eta} + \frac{\eta TG^2}{2}. \tag{41}$$

For the Coupling Error, using the Lipschitz property ($\|\nabla f_t(\mathbf{s}_t)\| \leq G$) and the geometric distribution of the staleness $(t - \tau(t))$ with mean $1/p$:

$$\mathbb{E}[\langle \nabla f_t(\mathbf{s}_t), \mathbf{s}_t - \mathbf{w}_t \rangle] \leq G\,\mathbb{E}[\|\mathbf{s}_t - \mathbf{w}_t\|] \leq \frac{\eta G^2}{p}. \tag{42}$$

Summing over $T$ gives $\mathbb{E}[\text{Coupling Error}] \leq \frac{T\eta G^2}{p}$. Combining the bounds, the total expected regret is upper bounded by:

$$\mathbb{E}[\mathsf{Reg}_T] \leq \underbrace{pT\lambda D}_{\text{Movement Cost}} + \underbrace{\frac{D^2}{2\eta} + \frac{\eta TG^2}{2}}_{\text{Virtual OGD Regret}} + \underbrace{\frac{T\eta G^2}{p}}_{\text{Coupling Error}} . \tag{43}$$

We are given a switching budget $M$. To ensure the expected number of switches is at most $M$, we set the switching probability to:

$$p = \frac{M}{T}.$$

Substituting $p = M/T$ into (43):

$$\mathbb{E}[\mathsf{Reg}_T] \leq M\lambda D + \left(\frac{D^2}{2\eta} + \frac{\eta TG^2}{2}\right) + \frac{T^2\eta G^2}{M}.$$

Assuming the sparsity regime $M \ll T$, the term $\frac{T^2\eta G^2}{M}$ dominates $\frac{\eta TG^2}{2}$. We simplify the objective to balancing the Virtual Regret (specifically the $D^2/2\eta$ term) against the Coupling Error:

$$\text{Bound}(\eta) \approx M\lambda D + \frac{D^2}{2\eta} + \frac{T^2\eta G^2}{M}.$$

Minimizing the $\eta$-dependent terms yields the optimal learning rate:

$$\frac{D^2}{2\eta^2} = \frac{T^2 G^2}{M} \implies \eta^* = \frac{D}{TG}\sqrt{\frac{M}{2}}.$$

Substituting $\eta^*$ back into the bound:

$$\frac{D^2}{2\eta^*} + \frac{T^2\eta^* G^2}{M} = \frac{D^2 TG}{2D\sqrt{M/2}} + \frac{T^2 G^2}{M}\frac{D}{TG}\sqrt{\frac{M}{2}} = \sqrt{2}\frac{DTG}{\sqrt{M}}.$$

Thus, the final bound is:

$$\mathbb{E}[\mathsf{Reg}_T] \leq M\lambda D + O\left(DG\frac{T}{\sqrt{M}}\right).$$

This completes the proof. $\qquad\square$

# F. Full Trigger Analysis

Table 3 provides the complete breakdown of update counts for the heterogeneous graph experiment in Section 7.2.

*Table 3.* **Full Trigger Analysis.** Breakdown of update counts by clique stiffness ($\lambda_{\mathrm{reg}}$). In the Global strategy, stable cliques (e.g., ID 9) are forced to match the update frequency of volatile ones (e.g., ID 4).

| Clique ID | Stiffness ($\lambda_{\mathrm{reg}}$) | Global Moves | Async Moves | Waste Factor |
|:---:|:---:|:---:|:---:|:---:|
| 4 | 0.19 (Volatile) | 41 | 15 | 2.7× |
| 2 | 0.25 | 41 | 14 | 2.9× |
| 5 | 0.67 | 41 | 14 | 2.9× |
| 6 | 0.86 | 41 | 13 | 3.2× |
| 1 | 1.06 | 41 | 13 | 3.2× |
| 7 | 1.11 | 41 | 13 | 3.2× |
| 8 | 1.26 | 41 | 13 | 3.2× |
| 3 | 1.28 | 41 | 14 | 2.9× |
| 9 | 1.40 (Stable) | 41 | 11 | **3.7×** |
| 0 | 1.61 (Stable) | 41 | 11 | **3.7×** |
| **Total** | - | **410** | **131** | **3.1×** |

