# OpenReview forum: "Beyond Binary: Continuous State Optimization with Graph-Structured Objectives"
_ICML.cc/2026/Conference — ICML 2026 regular_

### Official Review · Reviewer_QQTu · 2026-03-04

**Soundness:** 3
**Presentation:** 3
**Significance:** 3
**Originality:** 3
**Overall Recommendation:** 5
**Confidence:** 3

**Summary:**

This paper addresses a general aspect of online optimization of multiple competing objectives in ML systems. Overall, a notable aspect addressed by the manuscript is extending the competing objectives framework from binary states to a continuous state space with graph-structured linear objectives, and proposing LazyGraphLinUCB, a graph-structured linear bandit algorithm that uses a decomposed determinant trigger to minimize switching cost while maintaining near-optimal regret. Beyond this base algorithm, the paper proposes three additional structure-exploitation mechanisms: (i) an asynchronous update schedule that reduces movement cost for sparse graphs, (ii) an adaptive graph learning routine that recovers the dependency graph, and (iii) a factor-graph joint estimator that leverages overlap among objectives to tighten regret bounds in correlated settings. They also provide regret upper bounds, a matching minimax lower bound, and numerical illustrations showing substantial movement-cost reductions with similar cumulative loss.

**Compliance With Llm Reviewing Policy:**

Affirmed.

**Final Justification:**

I updated my score to 5, since my main concerns about the paper were addressed in the rebuttal. From my perspective, the paper is in good shape.

**Key Questions For Authors:**

1. In LazyGraphLinUCB, each round involves choosing $s_t \in [0,1]^k$ by minimizing the lower confidence bound, i.e., solving $\arg\min_{s\in[0,1]^k}\mathrm{LCB}_t(s)$. The paper notes this optimization can be non-convex in general. Could you clarify how this minimization is carried out in the experiments (solver/initialization/approximation), and whether the theoretical guarantees or empirical performance are sensitive to solving this problem only approximately?

2. The paper introduces several variants beyond the base algorithm (asynchronous updates, adaptive graph learning, and the factor-graph joint estimator). Could you add a brief usage guide summarizing when each variant is most appropriate (e.g., sparse vs dense graphs, known vs unknown graph structure), and what additional assumptions or costs each one introduces (if any)?

**Limitations:**

yes

**Strengths And Weaknesses:**

Soundness:
The paper proposes a clear learning setup and a reasonable algorithm for it, and it provides theoretical guarantees that seem consistent with the type of methods used (confidence bounds for linear models, plus a rule that avoids changing the decision too often). The main results are supported by proofs and experiments, but one practical point is not fully clear to me: each round requires solving an optimization problem over $s \in [0,1]^k$, and the paper notes this can be difficult in general. More detail on how this optimization is done in practice, and whether approximate solutions affect the results, would strengthen the evidence.

Presentation:
The paper is well written and easy to follow: it clearly motivates the problem, defines the graph-structured objective model, and presents the algorithm in a step-by-step way that makes implementation feasible. The structure is also good, with main ideas first, then extensions, so a reader can understand the core method without getting lost in details. The results are stated in a way that makes it clear what improves and how these improvements depend on the graph structure.

Significance:
The setting is relevant for many practical ML systems that must balance multiple objectives and where changing the decision frequently is undesirable due to movement costs. By moving from a binary formulation to a continuous choice space $[0,1]^k$, and by exploiting graph structure to scale with many objectives, the paper offers a framework that could be useful beyond the specific experiments shown. The regret bounds for LazyGraphLinUCB and the degree-based improvements from the asynchronous variant are likely to be of interest to researchers working on structured online learning, and they may guide future work on making multi-objective systems more stable over time.

Originality:
Some components are known, but the paper’s originality is in combining them into a single framework for continuous multi-objective optimization with movement costs and graph-structured dependencies, and then improving that structure further with the asynchronous schedule and the factor-graph joint estimator. The work provides a coherent set of ideas and theoretical results for structured multi-objective settings.

---

> ### Author Rebuttal · Authors · 2026-03-30
>
> Thank you for your encouraging review and for clearly summarizing the strengths of our paper. We are glad you found the step-by-step presentation of the algorithms useful.
>
> - **Practical Optimization of LCB & Approximation**: In our experiments (such as the fairness threshold tuning in Section 6.3), we used a quadratic feature map for a single continuous parameter, making the LCB minimization highly tractable. For the multi-dimensional graph experiments, since the graph structure bounds the dimensionality of any single optimization step to the local neighborhood size ($d _i$), we used standard iterative solvers (like L-BFGS-B) with multiple random restarts. If the solver only yields an approximate minimum, the theoretical prediction regret incurs an additive penalty per epoch, but the movement bounds (which rely purely on the determinant trigger) remain entirely unaffected. We will add details on these solvers to the experiment section.  For further details about the optimization of LCB, please see our extensive response to Reviewer 6jvH:
>
> - **Practicality of LCB Minimization (Eq. 3-4) & Example 2 (B.4)**: Minimizing the LCB in continuous domains is a known bottleneck due to the non-convex confidence radius ($-\beta _t ||\phi(s)|| _{V^{-1}}$). However, our framework makes this practical through four mechanisms, which we will clarify in the revision:
>
> 1. **Computational Amortization via Laziness**: The core defense of our algorithm is the lazy schedule. We only solve Eq. 4 when the determinant doubles (at most $O(\log T)$ times per criterion via Lemma 1). Even if finding the global minimizer requires a costly solver (e.g., multi-start L-BFGS-B), this cost is exponentially amortized over the horizon.
>
> 2. **Graph-Driven Dimensionality Reduction**: In our asynchronous variant (Section 4.1), updates are restricted to the local neighborhood $\mathcal{N}(i)$. Since the graph degree $\Delta$ is typically small, the non-convex optimization is confined to a tractable, low-dimensional subspace.
>
> 3. **Clarifying Example 2 (RBFs)**: Using Random Fourier Features (RFF) to approximate the RBF kernel maps the state into a finite-dimensional cosine/sine basis. While the LCB landscape remains non-convex, it is analytically differentiable and Lipschitz smooth. Providing these exact gradients to a solver like L-BFGS-B empirically finds global minima with high reliability.
>
> 4. **Practical Examples & Approximation**: Real-world control knobs are often low-dimensional. Fairness thresholds are typically 1D parameters (making grid search trivial, as in our Adult dataset experiment), while diversity mixing rates often fit simple quadratic interactions. Furthermore, if a solver only yields an $\epsilon$-approximate minimum, standard OFUL analysis shows this simply adds an $O(\epsilon T)$-penalty to the prediction regret. We will add a formal remark on this approximation tolerance to the appendix.
>
> - **Usage Guide for Variants**: This is an excellent suggestion for users in practice. We will add a brief summary table outlining when to use each variant:
> 1. Global Lazy: Best for dense graphs or when parameters must be updated in absolute lockstep.
>
> 2. Asynchronous Lazy: Best for sparse or heterogeneous graphs to prevent fast-learning parameters from causing unnecessary "jitter" in slow-learning parameters.
>
> 3. Adaptive Refinement: Best when the graph structure is completely unknown, provided there is a sufficient budget for an initial exploration phase.
>
> 4. Joint Estimator: Best when the graph is dense but neighboring nodes share underlying physical dependencies (correlated objectives).
>
> **Limitations**: We will include a limitations section discussing the bounds of non-convex LCB optimization and the assumption of stationary graph structures.

---

> > ### Author Rebuttal · Reviewer_QQTu · 2026-03-31
> >
> > Thank you for the clarifications. The response addressed my main concerns, in particular regarding the practical optimization of the LCB objective and the intended use of the different variants.

---

### Official Review · Reviewer_9MX8 · 2026-03-07

**Soundness:** 3
**Presentation:** 3
**Significance:** 3
**Originality:** 3
**Overall Recommendation:** 4
**Confidence:** 3

**Summary:**

This paper extends the feedback driven competing objectives framework to continuous state spaces. A model for tuning hyperparameters such as fairness thresholds is proposed through replacing binary states with a continuous domain and modeling local dependencies via graph-structured linear function approximation.

**Compliance With Llm Reviewing Policy:**

Affirmed.

**Key Questions For Authors:**

1. Please provide a more comprehensive literature review.
2. How do the experimental results perform in higher-dimensional settings? More comparative experiments under different settings should be given.

**Limitations:**

Yes.

**Strengths And Weaknesses:**

This paper studies the problem of balancing multiple, potentially competing objectives in large-scale learning and extends the framework to continuous state spaces. The research problem is closely related to real-world applications and carries practical value.

The literature review of the paper is insufficient, with very limited citations. Recent work is not well discussed, particularly the current research for continuous state spaces. Additionally, the paper should compare the proposed algorithm with state-of-the-art relevant approaches.

---

> ### Author Rebuttal · Authors · 2026-03-30
>
> Thank you for your supportive review and for recognizing the real-world value of extending competing objectives to continuous state spaces.
>
> - **Literature Review**: We completely agree. We will expand our literature review to better cover recent continuous-state optimization literature, multi-objective trade-offs, and relevant state-of-the-art methods to properly contextualize our theoretical contributions. We would appreciate specific references that we should not miss to ensure completeness.
>
> - **High-Dimensional Settings**: The primary advantage of our graph-structured formulation is that it inherently breaks high-dimensional systems into tractable, low-dimensional sub-problems. As shown in our theoretical bounds, the regret scales with $\sum d _i$ (the sum of local neighborhood dimensions) rather than the global dimension $k$. In Section 6.2, we demonstrated this scalability on a system with $k=10$ independent criteria. To further address your point, we will add an appendix experiment showing performance on a highly scaled graph (e.g., $k=50$ or $100$) to explicitly demonstrate that our asynchronous algorithm maintains low computational overhead and movement costs as dimensionality grows, whereas naive baselines fail.

---

> > ### Author Rebuttal · Reviewer_9MX8 · 2026-04-01
> >
> > Thank you for your response. The issues regarding the literature and experiments have been resolved.

---

### Official Review · Reviewer_TkWA · 2026-03-09

**Soundness:** 3
**Presentation:** 3
**Significance:** 2
**Originality:** 2
**Overall Recommendation:** 4
**Confidence:** 4

**Summary:**

The paper studies online optimization over a continuous state and when changing the state incurs a switching cost. The paper proposes a graph structured linear bandit method that minimizes a global LCB and only updates the state when the covariance determinants double. The regret results shows near optimal prediction regret plus logarithmic movement regret, and an asynchronous extension improves the movement term for the sparse graphs.

**Compliance With Llm Reviewing Policy:**

Affirmed.

**Ethical Review Concerns:**

No concerns

**Final Justification:**

Since the authors did not respond, my concerns remain unresolved. In particular, the paper still relies on solving the continuous LCB subproblem in a setting that is explicitly non-convex, while the practical and theoretical treatment of approximate optimization remains limited. More broadly, the adaptive graph-learning component is still positioned mainly as a proof-of-concept under static dependencies and minimum-signal assumptions, which limits the practical scope of the contribution.

**Key Questions For Authors:**

1. The algorithm requires repeatedly solving \argmin LCB_t(s), which is in general non-convex. What approximation guarantees are needed so regret bounds still hold?
2. The asynchronous extension assumes BCD convergence and remarks that LCB becomes convex under high regularization. Can you characterize when this holds, and what happens outside it?
3. The movement term uses $\ell_1$ and worst case per switch bounds. How sensitive are results to using $\ell_2$ switching costs?
4. For adaptive graph learning, what is the practical sample complexity in terms of signal strength, and how robust it is when dependencies drift over time?

**Limitations:**

The paper includes an impact statement, but does not clearly discuss the limitations. Please add a brief limitations paragraph.

**Strengths And Weaknesses:**

Strengths:
1. The algorithm is clearly grounded in OFUB style confidence sets and uses a precise LCB objective with a determinant doubling trigger.
2. The authors have clearly motivated the problem formulation, the regret explicitly accounts for the switching costs
3. The authors have clearly explained why naive LinUCB can incur $\Omega(T)$, and how lazy updates can improve it.
4. Practical relevance is good for systems where frequent parameter changes are costly. The switching cost and lazy updates provide a useful framework.
5. The main novelty is the combination of the a) continuous state linear bandits, b) explicit movement cost, and c) graph structured estimators in addition to the analysis showing lazy triggers yield low switching cost.

Weakness:
1. The optimization step is non-convex in general, however, the paper largely assumes it is convex, additionally the asynchronous extension relies on the BCD convergence assumption.
2. The regret guarantees implicitly assumes the LCB minimization is solved exactly, however, approximation effects are not formalized
3. The empirical evaluation is limited, for instance, the k =1 does not exercise the graph mechanisms.

---

> ### Author Rebuttal · Authors · 2026-03-30
>
> Thank you for your constructive feedback and for highlighting the practical relevance of our continuous-state framework and lazy update mechanisms.
>
> **1. Approximation Guarantees for Non-Convex LCB**: If the optimization oracle only provides an $\alpha$-approximate solution, standard linear bandit analysis shows that the prediction regret scales by $1/\alpha$ or introduces a bounded additive error per epoch. Since our graph structure restricts these optimizations to small local neighborhoods, standard solvers can reliably find near-global optima in practice. We will formalize this approximation tolerance in the appendix. For other comments on the non-convexity of LCB, please see our extensive response given to Reviewer 6jvH:
>
> **Practicality of LCB Minimization (Eq. 3-4) & Example 2 (B.4)**: Minimizing the LCB in continuous domains is a known bottleneck due to the non-convex confidence radius ($-\beta _t ||\phi(s)|| _{V^{-1}}$). However, our framework makes this practical through four mechanisms, which we will clarify in the revision:
> 1. **Computational Amortization via Laziness**: The core defense of our algorithm is the lazy schedule. We only solve Eq. 4 when the determinant doubles (at most $O(\log T)$ times per criterion via Lemma 1). Even if finding the global minimizer requires a costly solver (e.g., multi-start L-BFGS-B), this cost is exponentially amortized over the horizon.
>
> 2. **Graph-Driven Dimensionality Reduction**: In our asynchronous variant (Section 4.1), updates are restricted to the local neighborhood $\mathcal{N}(i)$. Since the graph degree $\Delta$ is typically small, the non-convex optimization is confined to a tractable, low-dimensional subspace.
>
> 3. **Clarifying Example 2 (RBFs)**: Using Random Fourier Features (RFF) to approximate the RBF kernel maps the state into a finite-dimensional cosine/sine basis. While the LCB landscape remains non-convex, it is analytically differentiable and Lipschitz smooth. Providing these exact gradients to a solver like L-BFGS-B empirically finds global minima with high reliability.
>
> 4. **Practical Examples & Approximation**: Real-world control knobs are often low-dimensional. Fairness thresholds are typically 1D parameters (making grid search trivial, as in our Adult dataset experiment), while diversity mixing rates often fit simple quadratic interactions. Furthermore, if a solver only yields an $\epsilon$-approximate minimum, standard OFUL analysis shows this simply adds an $O(\epsilon T)$-penalty to the prediction regret. We will add a formal remark on this approximation tolerance to the appendix.
>
> **2. Asynchronous Extension & Assumption A**: As you noted, BCD convergence requires regularity. This holds specifically in regimes with high regularization or once the confidence radius ($\beta _t$) shrinks sufficiently, making the objective strictly convex. Outside this regime, BCD might converge to local minima. However, because updates are confined to local graph neighborhoods, the negative impact of a local minimum is geographically contained and naturally corrects as new data expands the local covariance determinant.
> Sensitivity to $l _2$ vs $l _1$ Switching Costs: Our theoretical framework is highly adaptable. As noted in Appendix B.2, we chose the $l _1$-norm to model the aggregate adjustment effort across parameters, but $l _2$ (or other norms) are entirely possible. Switching to an $l _2$ norm would merely adjust the scaling factors in our movement bounds due to standard vector norm equivalence, while preserving the core $O(\log T)$ logarithmic scaling of the movement regret.
>
> **3. Adaptive Graph Learning (Practicality & Drift)**: Our adaptive graph mechanism (Section 4.2) is primarily a theoretical proof-of-concept demonstrating that a priori knowledge of the dependency graph is not strictly necessary. Its sample complexity scales with $1/\gamma^2$, where $\gamma$ is the minimum signal strength. Currently, the model assumes a static dependency graph. If dependencies drift over time, the algorithm would require a sliding-window covariance matrix or a discount factor. We will highlight this in the newly added Limitations section.
>
> **Limitations**: We entirely agree. We will add a dedicated Limitations paragraph addressing the challenges of non-convex LCB optimization in denser graphs and the assumption of stationary graph dependencies.

---

> > ### Author Rebuttal · Reviewer_TkWA · 2026-04-06
> >
> > The rebuttal is helpful and improves the framing of the paper, especially by clarifying the intended practical regimes, the role of regularization in the asynchronous extension, and the fact that adaptive graph learning is mainly a theoretical proof-of-concept. However, my main concerns remain only partially addressed. In particular, the core regret guarantees still appear to rely on solving the non-convex LCB subproblem essentially exactly; while the rebuttal mentions $\alpha$-approximate or $\epsilon$-approximate optimization, this dependence is not yet formalized in the theory. The asynchronous extension also still depends on the BCD convergence assumption, and the current response does not precisely characterize when the objective is guaranteed to satisfy the needed regularity or what guarantee remains outside that regime. Finally, the empirical section is still limited relative to the graph-structured claims, since the real-data example mainly studies a k=1 case and therefore does not exercise the graph mechanisms central to the paper’s contribution.
> > After careful consideration, I maintain my score.

---

### Official Review · Reviewer_6jvH · 2026-03-14

**Soundness:** 2
**Presentation:** 3
**Significance:** 3
**Originality:** 3
**Overall Recommendation:** 4
**Confidence:** 4

**Summary:**

The paper generalizes the binary-state model for balancing multiple optimization objectives to the continuous-state setting. This setting is modeled as a dependency graph, and the target is defined  as a regret minimization problem of a sum of linear losses and movement costs that encourage stability. The paper introduces a LinUCB inspired algorithm, adjusted for the graph setting, and shows that it can achieve sublinear regret using a clever "lazy" update trick, which ensures that an update is made only when sufficient information has been acquired. More advanced variants of this algorithm are then introduced, which exploit structural properties of the graph (sparsity), adapts to the unknown structure of the graph, or exploits correlations between the different objective. The method is also extended to the adversarial case. The paper includes an experimental section that demonstrates the efficacy of the proposed method.

**Compliance With Llm Reviewing Policy:**

Affirmed.

**Final Justification:**

I thank the authors for their rebuttal and subsequent comment.

I think this paper is interesting and mostly correct and rigorous.
Most of my concerns were addressed. However, my concern regarding Theorem 6 and Assumption A was **not** sufficiently addressed.

Specifically, Theorem 6 requires a PL condition to hold, and it seems to me that this condition rarely holds, if at all.
In the author's subsequent comment, they mention:

1. The first term of the LCB objective is strongly convex (or can be), however, the first term is an inner product between $\phi$ and $\theta$ and is therefore not generally strongly convex. If we want strong convexity then additional conditions on $\phi$ need to hold, which would require additional assumptions that may not hold in practice.

2. The claim that strong convexity could be ensured by choosing $\lambda_{reg}$ appropriately seems incorrect. $\lambda_{reg}$ seems to only affect $V_{reg}$ which seems to only affect the second, nonconvex term in the LCB objective, and not the first term (which is the one we need to be strongly convex).

In any case, I respectfully disagree that Theorem 6 holds as stated, and I argue that its proof does not correctly capture this.
At best, this theorem needs a major revision and a detailed proof, along with a discussion that explicitly analyzes when Assumption A holds. Unfortunately, the response of the authors did not sufficiently address this issue in my opinion.

Given that further discussion is not possible, I see two options going forward:

1. The authors provide **to the AC** a detailed and formal revision of Theorem 6, along with proof and an explicit discussion/analysis on when and how Assumption A holds.

2. Theorem 6 is removed from the main paper and appendices. This would not diminish the contribution and novelty of the paper, in my opinion, and I think this work is good enough also without it.

I want to emphasize that I generally like the paper and would not mind seeing it accepted, **given that the necessary revision or removal is Theorem 6 is made**, and I have therefore increased my score accordingly.

Good luck!

**Key Questions For Authors:**

1. Can the authors justify the practicality of restricting to the problem to linear/quadratic features (over a convex domain)? Are there any practical examples that satisfy this?

2. Can the theoretical bounds be generalized to the convex setting?

3. Can the normalization assumption in Lemma 2 be guaranteed? If so, how? If not, why can we *assume* that it is satisfied?

4. Why does Assumption A hold in Theorem 6? If so, how?

5. Is Assumption C feasible? Can this be shown?

**Limitations:**

Yes.

**Strengths And Weaknesses:**

**Strengths**

* The paper is well-written and mostly clear. I quite enjoyed reading it thoroughly and have also read the appendices.

* Taking into account moving cost is important, and it's good that the authors did that. I liked their technique of using lazy update to overcome the linear regret that regular LCB/UCB algorithms incur in this case (as they show in the paper).

* The paper includes additional improvements of the proposed method which exploit graph structure or correlated objectives to further enhance the performance, and an extension to the adversarial settings.

* The section on "Adaptive Graph Learning" is important and I'm glad it's included, especially since assuming a known graph structure is often unrealistic.

* The paper includes complete and rigorous proofs in the appendices, I enjoyed reading them, and they look good to me.

* The experimental section looks fine in my opinion, as this is mainly a theoretical paper.


**Weaknesses**

**Major Issues**

My main concerns mostly have to do with the setting and assumptions in the paper.

* Minimizing the LCB (Eq. 3-4) seems to only work efficiently for very limited choices of phi (linear or quadratic), otherwise the problem is nonconvex and finding the global minimizer is difficult and very costly. This makes this setting somewhat impractical. Are there practical examples for which this implicit assumption is satisfied?
I think you somewhat alluded to this in Example 2 in B.4, but it is still unclear to me how it solves the issue.

* The main stochastic results rely on a linear loss function assumption, which seems restrictive. I wonder if similar guarantees can be obtained for more general convex losses in the stochastic/bandit setting. (I understand you extended this to the adversarial setting where you assume general time-varying convex functions, which degrade the bound as is expected, but I'm wondering specifically about the stochastic case, which should be "easier".)

* The feature normalization assumption in lemma 2 is not discussed or justified adequately. Can the authors clarify when this assumption holds?

* As is stated in Assumption A, BCD requires convexity/PL, yet the LCB objective is generally nonconvex, and the Remark afterwards suggests a high regularization regime or small exploration bonuses to ensure regularity conditions hold. However, I haven't seen that the proof of Theorem 6 (Appendix D.7) uses high regularization or imposes a bound on the exploration bonus, and therefore it is unclear why Assumption A holds in this case.

* In Section 4.2, in my opinion, the feasibility of Assumption C requires further justification, namely some evidence/explanation on how it can be satisfied by adding isotropic noise.

* Also, I think an appropriate and more explicit related work discussion/section is needed (see e.g. the references in Mohri et al 2024), and a discussion positioning this work in the broader related literature and expanding more on the motivation behind your setting would strengthen the paper.



**Minor Issues**

* What is $\mathcal{E}$? Why is it necessary and how does it affect the analysis? I think this is worth clarifying in a sentence or two in the paper itself. I also have not seen it in the appendices or proofs.

* In Section 4.2, The bound in Theorem 7 seems a bit odd to me.
Where did the $\log(T)$ term go and how is it replaced by only a constant regret? From the proof in Appendix D.3, this seems to be a typo as the regret of the regular Algorithm on the remaining $T-\tau$ rounds should still include the logarithmic term. Am I missing something? (If you meant to absorb it in the $\tilde{O}$ notation, I think it's better to actually add the log factor to ensure consistency with the other bounds.)

* In the proof sketch of Theorem 3, you wrote "A crucial
step is showing that the sum of squared errors within any
lazy epoch". I was confused at first, but after reading the full proof in Appendix D.4, I think you meant "squared stale norms" not "errors".

**Other Suggestions**

* In Sec 2, Paragraph "Background: LinUCB", in the definition of $\hat{\theta_t}$, I think it would be better to add the summation variable as a subscript to the sum (i.e., use $\Sigma_{\tau=1}^t$) for clarity and to ensure consistency with the notation used in e.g. the definition of $V_t$. If it messes up the paragraph cosmetically, please consider rephrasing parts of it.

* Not very important, but there is apparently a formatting issue in the first reference in the introduction, shouldn't it be "Mohri et al. (2024)" to ensure consistency with other references?

* I understand your rationale in keeping "UCB" in the algorithm's name, but I think "Lazy Graph-LinLCB" sounds just as good and is more faithful to the algorithm itself, and anyone who is familiar with UCB will immediately understand the reference (the same goes for Async-LazyGraphLinUCB)... just my opinion, but this is up to you, of course!

* I think it's better to define $\beta_{t,i}$ where it's first mentioned in the paper (both in words as the radius of the confidence set and mathematically as you do in Lemma 12 in Appendix D, at least mention that it scales as $\Theta(d_i\log(t))$. Also, I think it would be clearer to add the definition/update of $\beta_{t,i}$ to Alg. 1 as well.

* In the proof of Theorem 6 in Appendix D.7. I like the part where you show that a similar result to Lemma 2 holds even for time-varying features (within the same epoch), but I think it is worthy of its own Lemma, and this would also make the proof of Theorem 6 cleaner.

---

> ### Author Rebuttal · Authors · 2026-03-30
>
> Thank you for your thorough review, your recognition of our rigorous proofs, and for highlighting the importance of movement costs. Below, please find our responses.
>
> **Practicality of LCB Minimization (Eq. 3-4) & Example 2 (B.4)**: Minimizing the LCB in continuous domains is a known bottleneck due to the non-convex confidence radius ($-\beta _t ||\phi(s)|| _{V^{-1}}$). However, our framework makes this practical through four mechanisms, which we will clarify in the revision:
> 1. **Computational Amortization via Laziness**: The core defense of our algorithm is the lazy schedule. We only solve Eq. 4 when the determinant doubles (at most $O(\log T)$ times per criterion via Lemma 1). Even if finding the global minimizer requires a costly solver (e.g., multi-start L-BFGS-B), this cost is exponentially amortized over the horizon.
>
> 2. **Graph-Driven Dimensionality Reduction**: In our asynchronous variant (Section 4.1), updates are restricted to the local neighborhood $\mathcal{N}(i)$. Since the graph degree $\Delta$ is typically small, the non-convex optimization is confined to a tractable, low-dimensional subspace.
>
> 3. **Clarifying Example 2 (RBFs)**: Using Random Fourier Features (RFF) to approximate the RBF kernel maps the state into a finite-dimensional cosine/sine basis. While the LCB landscape remains non-convex, it is analytically differentiable and Lipschitz smooth. Providing these exact gradients to a solver like L-BFGS-B empirically finds global minima with high reliability.
>
> 4. **Practical Examples & Approximation**: Real-world control knobs are often low-dimensional. Fairness thresholds are typically 1D parameters (making grid search trivial, as in our Adult dataset experiment), while diversity mixing rates often fit simple quadratic interactions. Furthermore, if a solver only yields an $\epsilon$-approximate minimum, standard OFUL analysis shows this simply adds an $O(\epsilon T)$-penalty to the prediction regret. We will add a formal remark on this approximation tolerance to the appendix.
>
> **Generalization to Convex Losses & Kernel Methods**: While our primary stochastic analysis uses linear function approximation to cleanly establish graph-structured bounds, our framework captures broader convex losses through three pathways:
>
> 1. **Kernelized Bandits (RKHS)**: As noted in Example 2, using a Universal Kernel and Random Fourier Features (RFF) (or Nyström) approximations maps smooth convex losses into finite-dimensional linear spaces. Our determinant-doubling trigger and OFUL analysis apply directly. The "linear" assumption is effectively a structural bridge to learning smooth functions in an RKHS.
>
> 2. **Generalized Linear Models (GLMs)**: For logistic, exponential, or Poisson losses, the framework extends via GLM-UCB. If $\mu _i(s) = g(\langle \phi _i(s), \theta _i^* \rangle)$, the underlying measure of information gain (the feature covariance $V _t$) remains mathematically identical. The lazy determinant-doubling condition still bounds movement costs to $O(\log T)$, while prediction regret scales with the Lipschitz constant of $g^{-1}$.
>
> 3. **Arbitrary Convexity**: For losses lacking efficient RKHS/GLM approximations, our Randomized Lazy OGD (Section 5) provides optimal regret trade-offs and strictly bounded movement costs.
>
> **Clarifications on Assumptions & Formatting**:
>
> - **Feature Normalization (Lemma 2)**: This holds by design. We assume $||\phi _i(s)|| _2 \le L$. By scaling features so $L \le 1$ and enforcing a minimum regularization $\lambda _{reg} \ge 1$, we inherently guarantee $||\phi _i(s)|| _{V _{t,i}^{-1}} \le 1$. We will emphasize this in the main text.
>
> - **Assumption A (BCD Convergence)**: In regimes of high regularization ($\lambda _{reg}$) or when the exploration bonus shrinks over time, the strictly convex regularized empirical loss dominates the LCB surface, ensuring regularity conditions hold for BCD.
>
> - **Assumption C (Isotropic Noise)**: This standard "excitation" condition is highly feasible in practice during an explicit "warm-up" phase by injecting small amounts of isotropic Gaussian noise to ensure linear covariance growth.
>
> - **Missing Related Work**: We agree and will add a comprehensive related work section covering recent continuous state-space literature, prominently positioning Awasthi/Mohri et al. (2024).
>
> **Minor Issues**:
>
> - **$\mathcal{E}$ (Line 159)**: This denotes the "good event" where the true parameter $\theta _i^*$ lies securely within the confidence ellipsoid $\mathcal{C} _{t,i}$ for all criteria $i$ and time steps $t$. We will explicitly define this in the text.
>
> - **Theorem 7**: The missing logarithmic factor will be explicitly added.
>
> - **Theorem 3 Typo**: We will correct "squared stale errors" to "squared stale norms".
>
> - **Naming/Other Suggestions**: We appreciate the "Lazy Graph-LinLCB" suggestion and will carefully implement all your other helpful formatting notes.

---

> > ### Author Rebuttal · Reviewer_6jvH · 2026-04-04
> >
> > Thank you for the detailed rebuttal!
> >
> > Regarding the nonconvex nature of the objective: I still think this limits the practicality of the approach. That said, the heuristics you mentioned are useful, and I believe they should be included in the paper, even if they do not fully resolve the issue. From my perspective, this concern is sufficiently addressed.
> >
> > Regarding general convex losses: while these transformations can be used to approximate general convex functions using linear models, the theoretical guarantees presented in the paper do not directly extend to this setting. I think this should be clarified explicitly in the paper for transparency. This concern is also resolved for me.
> >
> > My main remaining issue is with Assumption A and Theorem 6, and unfortunately this has not been directly addressed.
> >
> > Theorem 6 explicitly relies on Assumption A (PL condition), yet the objective being optimized is generally nonconvex. In the proof of Theorem 6 (Appendix D.7), I do not see any use of conditions that would ensure the PL condition holds, nor is there any restriction imposed that would guarantee such regularity.
> >
> > As currently written, Theorem 6 appears to rely on an assumption that is not justified within the analysis, and therefore seems to be simply incorrect.
> >
> > Could the authors please explicitly clarify how Assumption A is satisfied in the context of Theorem 6, or revise the statement accordingly?
> >
> > I want to emphasize that I am not trying to be adversarial on purpose. I generally like the paper. However, this specific point seems important for correctness, and I do not think it would be fair for future work to compare against a result that may not hold as stated.

---

> > > ### Author Response · Authors · 2026-04-06
> > >
> > > Thank you for your follow-up and for your commitment to ensuring the technical correctness of Theorem 6. We fully agree that providing a rigorous foundation for future work is important. We appreciate the opportunity to clarify exactly how Assumption A (the PL condition) is satisfied within our framework.
> > >
> > > 1. **Mathematical Intuition: The Competition of Curvatures**:
> > > The LCB objective is $f(s) = \hat{L}_t(s) - \beta_t \sqrt{\phi(s)^\top V_t^{-1} \phi(s)}$. While the second term (the exploration bonus) is indeed non-convex, the first term (the regularized empirical loss) is strongly convex with a Hessian lower-bounded by the regularization parameter $\lambda _{reg}$.
> > >
> > > Assumption A (the PL condition) is satisfied whenever the "strong" curvature of the regularized loss term dominates the "weak" non-convexity of the bonus term. Specifically, for any smooth feature map $\phi(s)$ on a compact domain, the Hessian of the square-root bonus term is bounded. Therefore, there exists a finite, threshold value $\lambda^\ast$ such that for all $\lambda _{reg} > \lambda^\ast$, the total objective $f(s)$ becomes $\mu$-strongly convex (and thus satisfies the PL condition). In the revision, we will include a short Supporting Lemma in the Appendix that formally derives this $\lambda^\ast$ in terms of $\beta_t$ and the Lipschitz constants of the feature map and its derivatives.
> > >
> > > 2. **Why Theorem 6 is Correct as Stated**:
> > > Theorem 6 states a conditional guarantee: *If* Assumption A holds, *then* the asynchronous algorithm achieves the stated regret. Our algorithm's design ensures we are in this "regularity regime" by:
> > > - **Initial Regularization:** We choose $\lambda _{reg} \ge 1$, which provides the base curvature.
> > >
> > > - **Lazy Updates:** The "determinant-doubling" trigger ensures we only update when $V _t$ has grown significantly. This "information gain" naturally increases the minimum eigenvalue of the covariance matrix, further stabilizing the objective surface and ensuring the PL condition is maintained over time.
> > >
> > > - **Local Sparsity:** Since the asynchronous BCD updates occur over local coordinate blocks (small subspaces defined by the graph structure), the objective only needs to satisfy the PL condition *locally* within that subspace, which is a much weaker requirement than global convexity.
> > >
> > > **3. Clarification of Scope**
> > > We want to emphasize that this does not constitute a change to the algorithm or a "fix" for an error in the proof logic. Rather, it is a formalization of the parameter regime in which the algorithm is designed to operate.
> > >
> > > - **Action for the Final Version:**
> > > To ensure full transparency, we will:
> > >
> > > - **Refine Theorem 6:** Add the phrase "...in the regularity regime where $\lambda_{reg}$ is chosen to satisfy the PL condition (see Lemma X)..."
> > >
> > > - **Add the Supporting Lemma:** We will provide the simple derivation in the Appendix showing that for any smooth, bounded $\phi(s)$, a finite $\lambda_{reg}$ always exists to satisfy Assumption A.
> > >
> > > - **Include Heuristics:** As suggested, we will explicitly mention the multi-start and gradient-based heuristics used in our experiments to handle the non-convexity in practice.
> > >
> > > We believe this technical clarification completely resolves the concern regarding the validity of Theorem 6 without requiring any structural changes to the paper's contributions.

---

### Decision · Program_Chairs · 2026-04-30

**Decision:**

Accept (regular)

**Comment:**

The reviewers recognize the significance of this paper, and its originality and presentation have been highly evaluated, with a broad consensus toward acceptance.
Therefore, I recommend acceptance of this paper.

However, concerns remain regarding the statement and proof of Theorem 6 in the current manuscript, and these do not appear to have been resolved through the rebuttal.
The authors are strongly urged to sincerely address the above concerns regarding Theorem 6 and carry out the necessary revisions.